# Patient-specific logic models of signaling pathways from screenings on cancer biopsies to prioritize personalized combination therapies

Federica Eduati[1,2,3,4] iD, Patricia Jaaks[5], Jessica Wappler[6], Thorsten Cramer[6,7,8] iD, Christoph A Merten[1], Mathew J Garnett[5] & Julio Saez-Rodriguez[2,3,9,*] iD

## Abstract

Mechanistic modeling of signaling pathways mediating patient-specific response to therapy can help to unveil resistance mechanisms and improve therapeutic strategies. Yet, creating such models for patients, in particular for solid malignancies, is challenging. A major hurdle to build these models is the limited material available that precludes the generation of large-scale perturbation data. Here, we present an approach that couples *ex vivo* high-throughput screenings of cancer biopsies using microfluidics with logic-based modeling to generate patient-specific dynamic models of extrinsic and intrinsic apoptosis signaling pathways. We used the resulting models to investigate heterogeneity in pancreatic cancer patients, showing dissimilarities especially in the PI3K-Akt pathway. Variation in model parameters reflected well the different tumor stages. Finally, we used our dynamic models to efficaciously predict new personalized combinatorial treatments. Our results suggest that our combination of microfluidic experiments and mathematical model can be a novel tool toward cancer precision medicine.

**Keywords** drug combinations; logic modeling; patient-specific models; precision oncology; signaling pathways

**Subject Categories** Cancer; Computational Biology; Signal Transduction

**Mol Syst Biol. (2020) 16: e8664**

## Introduction

Charting the dynamic wiring of signaling networks is of paramount importance to understand how cells respond to their environment. Identifying the differences in this wiring between normal and cancerous cells can shed light on the pathophysiology of tumors and pave the way for novel therapies (Werner *et al*, 2014; Saez-Rodriguez *et al*, 2015; Zañudo *et al*, 2018). A powerful tool to gain insight into these processes is to monitor the response of cells to multiple perturbations. When combined with mathematical modeling, such data can be used to determine cell type-specific wiring phenomena, predict efficacy of drug treatments, and understand resistance mechanisms (Saez-Rodriguez *et al*, 2011; Klinger *et al*, 2013; Merkle *et al*, 2016; Eduati *et al*, 2017; Hill *et al*, 2017).

Application of this strategy has been limited so far to *in vitro* contexts, as the experimental technologies to generate perturbation data require large amounts of material, which are unavailable from most primary tissues such as solid tumors. With recently developed organoid technologies, it became possible to generate large amounts of material *ex vivo*, enabling such screens in principle. However, they would be associated with large costs and, while recapitulating some of the features of the tumor physiology, the cells unavoidably diverge from the primary tumor as they are grown *ex vivo* (Letai, 2017). We have recently developed a novel strategy based on microfluidics that enables testing apoptosis induction upon a good number of conditions (56 with the current settings, with at least 20 replicates each) starting from as little as one million viable cells. Cells are encapsulated in 0.5 μl plugs together with an apoptosis assay and single or combined drugs. Using valves to control individual fluid inlets allows the automatic generation of plugs with different composition. These Microfluidics Perturbation Screenings (MPS) are suitable to collect such drug response datasets even with

1   European Molecular Biology Laboratory (EMBL), Genome Biology Unit, Heidelberg, Germany
2   European Molecular Biology Laboratory, European Bioinformatics Institute (EMBL-EBI), Hinxton, UK
3   Joint Research Centre for Computational Biomedicine (JRC-COMBINE), Faculty of Medicine, RWTH Aachen University, Aachen, Germany
4   Department of Biomedical Engineering, Eindhoven University of Technology, Eindhoven, The Netherlands
5   Wellcome Trust Sanger Institute, Hinxton, UK
6   Department Surgery, Molecular Tumor Biology, RWTH University Hospital, Aachen, Germany
7   ESCAM – European Surgery Center Aachen Maastricht, Aachen, Germany
8   ESCAM – European Surgery Center Aachen Maastricht, Maastricht, The Netherlands
9   Institute for Computational Biomedicine, Faculty of Medicine, BIOQUANT-Center, Heidelberg University, Heidelberg, Germany
    *Corresponding author. Tel: +49 6221 5451334; E-mail: julio.saez@bioquant.uni-heidelberg.de

the very limited number of cells available from tumor resection biopsies (Eduati et al, 2018).

In this study, we set out to construct cell type- and patient-specific models from the drug response data obtained using the MPS technology (overview of the pipeline in Fig 1A). We took advantage of our tool CellNOptR (Terfve et al, 2012), to train a general network of the underlying intrinsic and extrinsic apoptosis pathways from data obtained for two cell lines and biopsies from four pancreatic tumor patients at different stages. We obtained cell line- and patient-specific continuous logic models based on ordinary differential equations (ODEs). We found these models to be a useful tool to understand specific pathway deregulations and to predict new patient-specific therapies.

## Results

### Data and modeling of apoptosis pathways

Experimental data were generated using a novel Microfluidics Perturbation Screening (MPS) platform, which allowed performing combinatorial drug screening of biopsies from human tumors (Eduati et al, 2018; see Materials and Methods). Data represent caspase-3 (Cas3 in Fig 1B, marked in blue) activation after perturbation with 10 different compounds including seven kinase inhibitors (targeting IKKs, MEK, JAK, PI3K, EGFR, AKT, PDPK1—inhibited nodes are depicted in red in Fig 1B), one cytokine (TNF, stimulated node, in green), and two chemotherapeutic drugs (gemcitabine and Oxaliplatin). All 10 compounds were tested alone and in all 45 possible pairwise combinations (Fig 1C) on two pancreatic cancer cell lines (AsCP1 and BxPC3) and biopsies from four patients with pancreatic tumors at different stages (one intraepithelial neoplasia, two primary tumors, and one liver metastasis). Measurements were performed 16 h after perturbation, when Cas3 activation was shown to reach a plateau (Eduati et al, 2018).

To investigate the signaling mechanisms behind the differential drug responses of our cell lines and patients, we derived a general logic model of apoptosis pathways involved in the regulation of Cas3 (our measurement), which is considered as effector node and indicator of apoptosis. Models were then trained using the patient-specific experimental data from Eduati et al (2018) to obtain personalized models. The general model (Fig 1B) was built integrating information derived from literature and from public repositories (details in Materials and Methods section). The model describes both intrinsic (mediated by the mitochondria, named Mito in the model) and extrinsic (mediated by tumor necrosis factor receptors, TNFRs) apoptotic signals, including nodes encoding for both anti- and pro-apoptotic effects. We incorporated in the model all nodes perturbed by specific compounds in our screening such as targeted drugs (kinase-specific inhibitors) and the cytokine TNF. The effect of chemotherapeutic DNA damaging drugs was not included in the model since they inhibit DNA replication rather than acting directly on specific signaling nodes. However, nodes such as p53, which are activated by DNA damaging drugs, are included in the model since they are key elements of different pathways. Since our screening included two AKT inhibitors (i.e., MK-2206 and PHT-427) with different mechanisms of action (allosteric and PH domain inhibitors, respectively), they were modeled as acting on two different nodes

(AktM and AktP, respectively), both needed for the activation of AKT.

The logic model includes AND gates (circles in Fig 1B) when all upstream regulators are needed to activate a node, while cases with multiple independent regulators are considered as OR gates. The logic model is interpreted using the logic-based ordinary differential equation formalism (logic ODEs; Wittmann et al, 2009) as implemented in CellNOptR (Terfve et al, 2012). This formalism allows to maintain the simple causal structure of logic models, while considering also the dynamic nature of the interactions and the continuous scale for the activation of the nodes, by using ODEs. Thus, they are not limited to capture only binary events as is the case for Boolean models. As previously described (Eduati et al, 2017), we consider one parameter for each edge $j \rightarrow i$ in the network, which characterize the strength of the regulation of species $i$ dependent on species $j$ and one parameter for each node $i$, which represents the responsiveness of the node (see Materials and Methods).

### Calibration of the apoptosis model for cell lines

The parameters of the generic model were fitted separately to the data of each cell line, resulting in specific models tailored to the experimental data for each cell line (more details in Materials and Methods section). Parameter fitting was repeated 10 times, and performances were assessed using different metrics to compare model simulations with the experimental data, showing good and quite robust performances (average metrics for AsPC1 and BxPC3, respectively: Pearson correlation 0.72, 0.74; mean squared error 0.03, 0.02; coefficient of determination 0.5, 0.5; Appendix Fig S1A). Model simulations for the best specific models were compared with the corresponding measured experimental data, showing a very good agreement (Pearson correlation equal to 0.89 and 0.83 for AsPC1 and BxPC3, respectively; Appendix Fig S1B and C).

The calibrated models for these two cell lines were then used to uncover potential differentially regulated mechanisms which are behind the different drug responses of the cell lines. Due to the limited number of data and the complex nature of the signaling pathways involved in the activation of apoptosis, not all model parameters can be estimated with the same confidence. In order to estimate the variability of the optimized parameter values, we derived a bootstrapped distribution for each parameter for each cell line, by repeating the optimization 500 times while randomly resampling the data with replacement (Dataset EV1). Results from bootstrap iterations were used to assess whether any specific experiment was essential for constraining the model (Appendix Fig S2) and for making predictions (Appendix Fig S3), by considering the left out experiments (due to resampling with replacement) as validation set. This analysis confirmed that even if a specific condition is missing from the training set, this does not significantly affect the model and the resulting predictions. This is probably due to the fact that individual drugs are used in multiple experiments (all the combinations); therefore, even if one specific condition is missing from the training set, the model is still constrained by all the other conditions. These bootstrap distributions were then used to compare the two cell lines using statistical tests to highlight significant differences (Wilcoxon sum rank test, adjusted $P$-value < 0.01, effect size > 0.2, see Materials and Methods) as represented in Fig 2A. The comparison revealed some regulatory mechanisms which are

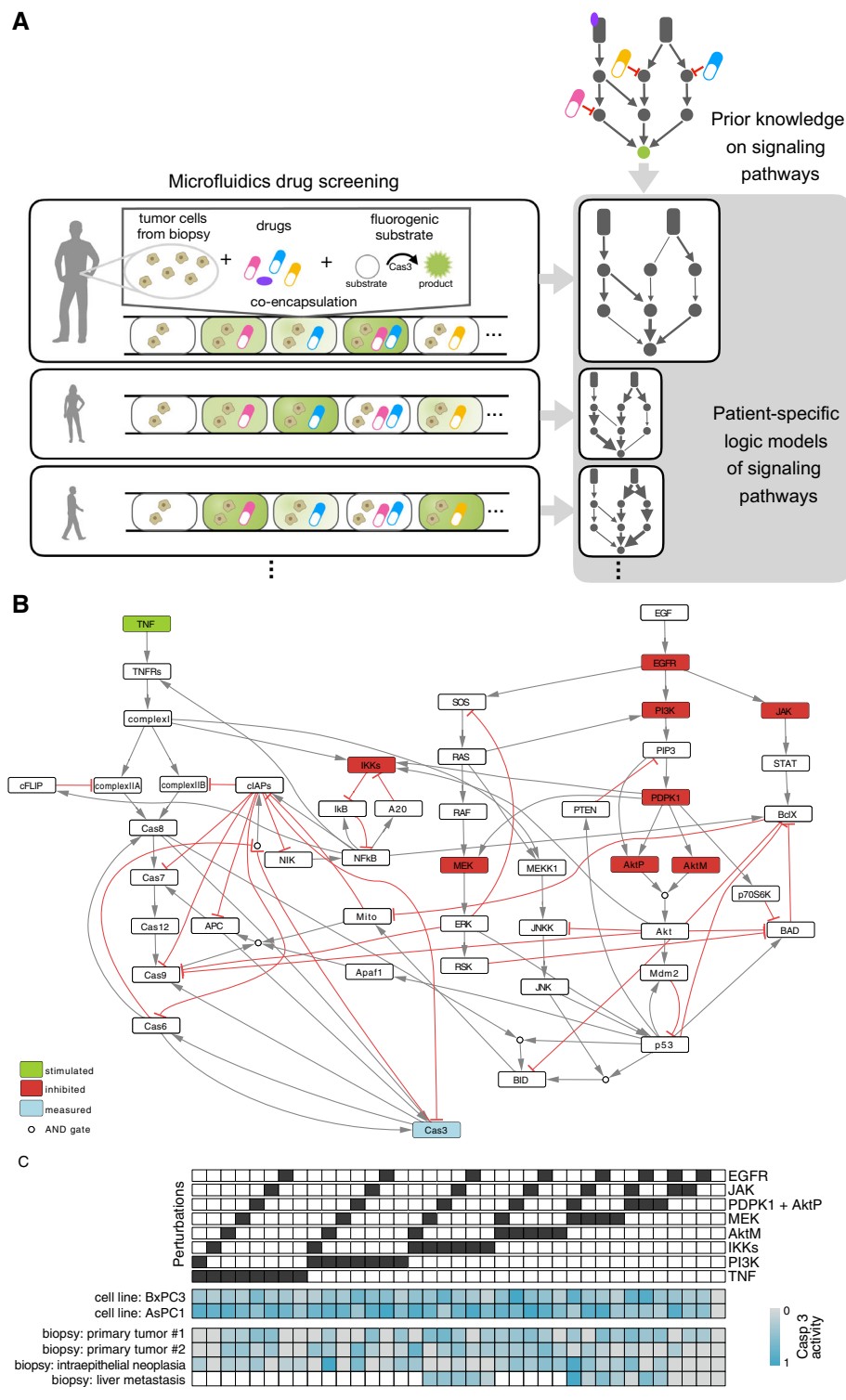

**Figure 1. Apoptosis pathways and experimental data.**

A   Overview of the pipeline: Patient-specific mathematical models are built for each patient from the combinatorial Microfluidics Perturbation Screening (MPS) data measured on live cells from cancer patient biopsies.

B   Logic model of intrinsic and extrinsic apoptosis pathways regulating Cas3 (our readout, blue node), including all nodes which are perturbed (stimulated in green, inhibited in red) in the experiments.

C   Experimental data consisting of 37 different experimental conditions (columns) for six samples (rows; 2 cell lines and 4 biopsies).

upregulated in either AsPC1 or BxPC3. Additionally, the dynamic of Cas3 activation appears to be faster in BxPC3 (node border in purple). Main differences involve the PI3K-Akt pathway. The main linear pathway is more active in BxPC3, whereby the negative feedback loop, from p53 to PIP3 mediated by PTEN, is stronger in AsPC1. These differences in the model parameters cause changes in the dynamic behavior of the system (Fig 2B) and are behind the differential activation of Cas3 in response to drugs.

We then investigated if these differences in dynamic behavior could be derived from their genetic makeup (Garcia-Alonso et al,

2018). From the proteins in our model, only KRAS is functionally mutated in AsPC1 and TP53 in BxPC3. Furthermore, no known direct regulators of the nodes in our model—from those in Omni-Path (Türei et al, 2016), a compendium of pathway resources—were mutated. Interestingly, KRAS and TP53 are indeed involved in the pathways that we found to be differentially activated between the two cell lines, but on their own, they cannot explain the differences in pathway structure in terms of strength of regulations. Therefore, information on mutations alone would not be sufficient to describe the dynamics of the pathways that mediate apoptosis upon drug

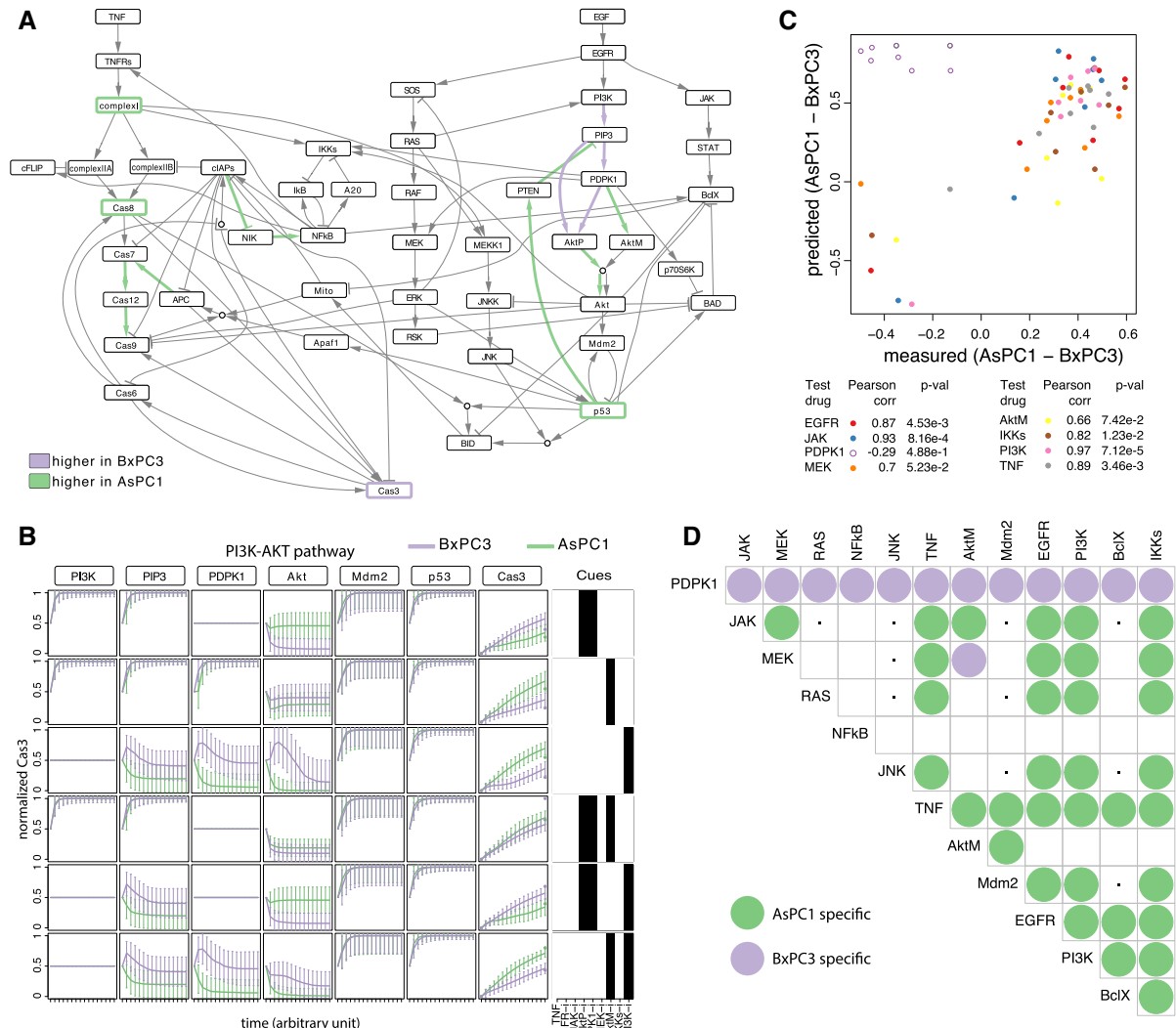

**Figure 2. Models, predictions, and validation for AsPC1 and BxPC3 cell lines.**

A  Differentially regulated mechanisms in AsPC1 or BxPC3, highlighted in green or purple depending on whether the corresponding estimated parameters are higher in AsPC1 or BxPC3, respectively.

B  Time course simulation of PI3K-AKT pathway and related inhibitors in AsPC1 and BxPC3. Lines represent median values, and error bars represent standard deviation from the bootstrapped simulations.

C  Assessment of model predictions by cross-validation, removing for each repetition all experiments involving one of the eight drugs (the corresponding drug targets are reported in the legend) from the training set and using it as test set. Bad predictions (Pearson Correlation < 0.6—which corresponds to removal of drug targeting PDPK1) are marked with empty dots.

D  New drug combinations predicted to be highly specific for each cell line, limited to targettable nodes. Combinations predicted to be specific for AsPC1 are marked in green and those specific for BxPC3 in purple. Combinations predicted to have no effect in both cell lines are marked with a dot.

treatment. The same holds when looking at basal transcriptomics (Iorio et al, 2016; Appendix Fig S4), supporting the observation that static data are not sufficient to investigate the dynamics of a complex system. Differences in signaling pathways (Fig 2A) are driven by the functional status of the nodes that cannot be inferred solely from differences in gene expression (Appendix Fig S4).

## Model predictions and validation

We decided to test the predictive power of our cell line-specific mathematical models in two ways: (i) using cross-validation on the existing dataset and (ii) predicting the effect of new drug combinations that can be experimentally tested.

First, optimization was repeated randomly selecting eight conditions as validation set and using the remaining 36 for training (bootstrapping 100 times). This procedure was repeated for 20 randomized test sets, and results show a good correlation between the predictions and the measurements in the validation set (average Pearson Corr = 0.7). Even when the cross-validation was repeated removing all the experiments involving a specific drug each time (instead of random ones; Fig 2C), predictions are still very good (Pearson Corr range 0.66–0.97) for all drugs except PHT-427 (Akt and PDPK1 inhibitor, Pearson Corr = −0.29). This implies that experiments with PHT-427 are essential to define the models.

Mathematical models were then used to simulate the effect of targeting all nodes of the network in pairwise combinations, therefore simulating the effect of potential new drugs acting on pathways that were not previously tested using MPS. For each cell line, we simulated the effect of 186 new perturbations (12 on individual nodes and 174 on node pairs), by inhibiting the corresponding node in the model. Varying confidence of model parameter estimation from the available data is expected to affect the ability to predict certain conditions. By using the family of models optimized using bootstrap, we obtained a distribution of the predicted activation of Cas3 in response to the different simulated conditions, therefore retaining information on the confidence we have for each prediction. In particular, we identified the predictions which were significantly different between the two cell lines (Wilcoxon sum rank test, adjusted $P$-value < 0.01, effect size > 0.3, see Materials and Methods; Appendix Fig S5), focusing on those that involve the inhibition of pairwise combinations of the 12 targettable nodes (JAK, MEK, RAS, NFkB, JNK, TNF, AktM, Mdm2, EGFR, PI3K, BclX, IKKs; Fig 2D). This results in 66 drug combinations, 34 of which are specific for AsPC1 and 13 for BxPC3 and 10 have no effect on both cell lines.

We then tested experimentally three of the top combinatorial therapies predicted to be highly specific for one of the cell lines based on our mathematical models (Fig 3A–C). For all three cases, we had concordance between model prediction and experimental validation. Interestingly, all three combinations are targeting the PI3K/Akt pathway, which is the one where most differences were highlighted between the two cell lines. As the KRAS oncogene is mutated in > 95% of pancreatic adenocarcinomas (Jones et al, 2008), inhibition of signaling pathway downstream of KRAS is considered to be an attractive therapeutic approach. Against this background, we tested the combination of the Akt inhibitor MK-2206 with a MEK inhibitor (trametinib). As predicted by our model, the validation experiments showed that the combination of trametinib and MK-2206 is more efficacious and synergistic (based on

Bliss independence model) in BxPC3 than in AsPC1 cells (Fig 3D, Appendix Fig S6). In line with this, a recent report on preclinical models of pancreatic cancer displayed enhanced efficacy of gemcitabine plus nabPaclitaxel when combined with MK-2206 and trametinib (Awasthi et al, 2019). KRAS-mutant colorectal cancer was found to be extremely sensitive to combined inhibition of Bcl-2/Bcl-xL and mTORC1/2 (Faber et al, 2014). Interestingly, the combination of navitoclax (a Bcl-2/Bcl-xL/Bcl-W antagonist) with PHT-427 (targeting PDPK1 and Akt, the bona fide downstream effector of mTORC2) was predicted by our model to be specific for BxPC3 cells. Additionally, loss of function in PTEN was shown to be important for synergistic interaction between MEK and mTOR inhibitors (Milella et al, 2017). While both AsPC1 and BxPC3 are PTEN wild type, our model identified a weaker negative feedback loop mediated by PTEN in the BxPC3 cell line that could justify the observed synergy. This prediction was also confirmed by the experimental data showing that the combination of navitoclax and PHT-427 is more efficacious and synergistic in BxPC3 than in AsPC1 cells (Fig 3E, Appendix Fig S7). Of note, the combination of Bcl-2/Bcl-xL/Bcl-W antagonists and drugs targeting PDPK1/Akt has thus far not been discussed as a treatment option for pancreatic cancer, further demonstrating the applicability of our model to identify innovative combinatorial approaches. Finally, we tested the combination of a PI3K inhibitor (taselisib) with the Bcl-2/Bcl-xL/Bcl-W antagonist navitoclax. Agents targeting the PI3K pathway in combination with a Bcl-2 family inhibitor have been previously suggested to be relevant in the context of pancreatic cancer (Tan et al, 2013). Hence, being able to predict the efficacy of this combination for specific patients (or cell lines in this case) would be highly desirable. Our validation experiments showed that the combination of taselisib and navitoclax is more efficacious and synergistic in AsPC1 than in BxPC3 cells (Fig 3F, Appendix Fig S8), confirming our model-based predictions.

Encouraged by these *in vitro* results, we performed *in vivo* validation on xenograft mouse models (Dataset EV2 and EV3). As predicted by our mathematical models, when treated with a combination of trametinib and MK-2206, both mouse models showed a significantly different response ($P$-value = 0.04, see Materials and Methods for details) with BxPC3-derived mice showing a stronger response (Fig 3G). For the other treatments, no significant difference was found ($P$-value = 0.27 for navitoclax + taselisib, $P$-value = 0.22 for navitoclax + PHT-427). Additionally, we tested the effect of gemcitabine, the standard of care in pancreatic cancer, which showed a similar effect in both out mouse models ($P$-value = 0.75). Interestingly, the combination of trametinib and MK-2206 provides significantly better treatment with respect to gemcitabine in the BxPC3 mouse model ($P$-value = 0.04), while this is not the case for the AsPC1 mouse model ($P$-value = 0.77; Fig 3H). Overall, these results suggest that our mathematical models have a potential for predicting personalized treatment that can specifically improve *in vivo* response with respect to the standard of care.

## Personalized apoptosis models for patients' tumors

The same fitting pipeline previously described for the cell lines was applied to the data from the four pancreatic tumor biopsies (intraepithelial neoplasia, two primary tumors and one liver metastasis) to obtain personalized models. Patient-specific parameter distributions

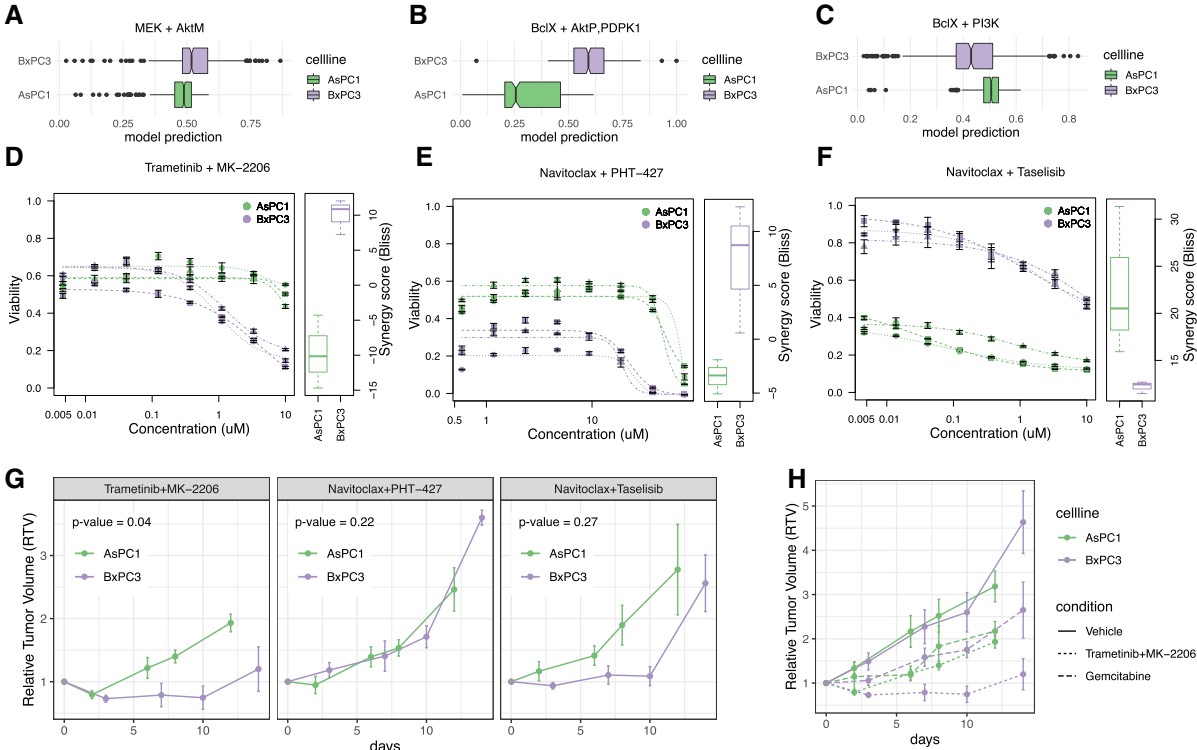

**Figure 3.** *In vitro* and *in vivo* experimental validation of model predictions.

A–C  Model simulations when inhibiting (A) MEK and AktM nodes, (B) BclX, AktP and PDPK1 nodes, (C) BclX and PI3K nodes. Data are shown using notched boxplots: the middle line represents the median, the box limits correspond to the interquartile range and the whiskers extend to the most extreme data point, which is no more than 1.5 times the length of the box away from the box (outliers are represented as dots).

D–F  *In vitro* experimental validation of the combination of (D) trametinib (MEK inhibitor, anchor drug at 1 μM) and MK-2206 (Akt inhibitor, 8-points 1:3 dilution series), (E) navitoclax (BclX inhibitor, anchor drug at 10 μM) and PHT-427 (AktP and PDPK1 inhibitor, 8-points 1:2 dilution series), (F) navitoclax (BclX inhibitor, anchor drug at 2.5 μM) and taselisib (PI3K inhibitor, 8 time points 1:3 dilution series). Data shown are for three biological replicates with three technical replicates each (error bars represent standard error of the technical replicates). Corresponding boxplots show the resulting synergy scores (Bliss model) computed for each biological replicate considering all concentrations of the anchor drug and the highest two concentrations of the combined drug. Summary statistics are represented using a horizontal line for the median and a box for the interquartile range. The whiskers extend to the most extreme data point, which is no more than 1.5 times the length of the box away from the box.

G  *In vivo* validation on cell line derived xenograft mouse models comparing the effect on the mouse models derived from the two cell lines. Data shown are for four mice (error bars represent the standard error—full data are provided as Dataset EV2 and EV3). *P*-values were derived using linear mixed-effect models to compare longitudinal data (corrected for multiple comparisons using Holm method).

H  *In vivo* comparison of the combination of trametinib and MK-2206 with the control condition (vehicle alone) and with the standard of care gemcitabine (error bars represent the standard error—full data are provided as Dataset EV2 and EV3).

were used to investigate patient heterogeneity at the level of mechanisms involved in apoptosis signaling pathways. Results are summarized in Fig 4A, showing the 16 (out of 93) parameters which are different in at least one patient (Kruskal–Wallis rank sum test, adjusted *P*-value < 0.01, effect size > 0.2, see Materials and Methods). For these parameters, we also performed post hoc pairwise statistical tests to directly compare all patients (Wilcoxon sum rank test, adjusted *P*-value < 0.01, effect size > 0.2, see Materials and Methods). For instance, for the parameter representing the EGFR → JAK regulation, the null hypothesis of equal distribution is not rejected when comparing the two primary tumors (lower two boxes in gold) between themself and with respect to intraepithelial neoplasia (top-left box, half gold next to the corresponding interaction in Fig 3A); however, it is rejected when comparing each of them with liver metastasis (top-right box, in cyan). Also, the comparison of intraepithelial neoplasia and liver metastasis suggests that the two samples do not come from different distributions (top boxes, both cyan).

Overall, the most different sample is the liver metastasis (different from all the others in 7 of the 16 heterogeneous parameters (44%)), especially in the extrinsic apoptosis pathway mediated by complex I, cIAPs, and Cas8. This larger dissimilarity could be justified by the difference both in stage and in tissue, since all other samples were resected from the pancreas. Also, the intraepithelial neoplasia shows a quite high level of dissimilarity (37.5%), localized in particular in the IKKs-NFkB pathway, which could reflect the more advanced stage of the disease. Interestingly, the two primary tumors are the most similar to each other (similar in 11 out of the 16 parameters, corresponding to 69%). However, significant differences were found especially in the PI3K-AKT pathway, similarly to what we observed for the two pancreatic cancer cell lines. Importantly, these similarities, which reflect the different tumor stages, were not evident directly from the data, where primary tumor #1 clusters closer to the liver metastasis and primary tumor #2 closer to the intraepithelial neoplasia (Appendix Fig S9).

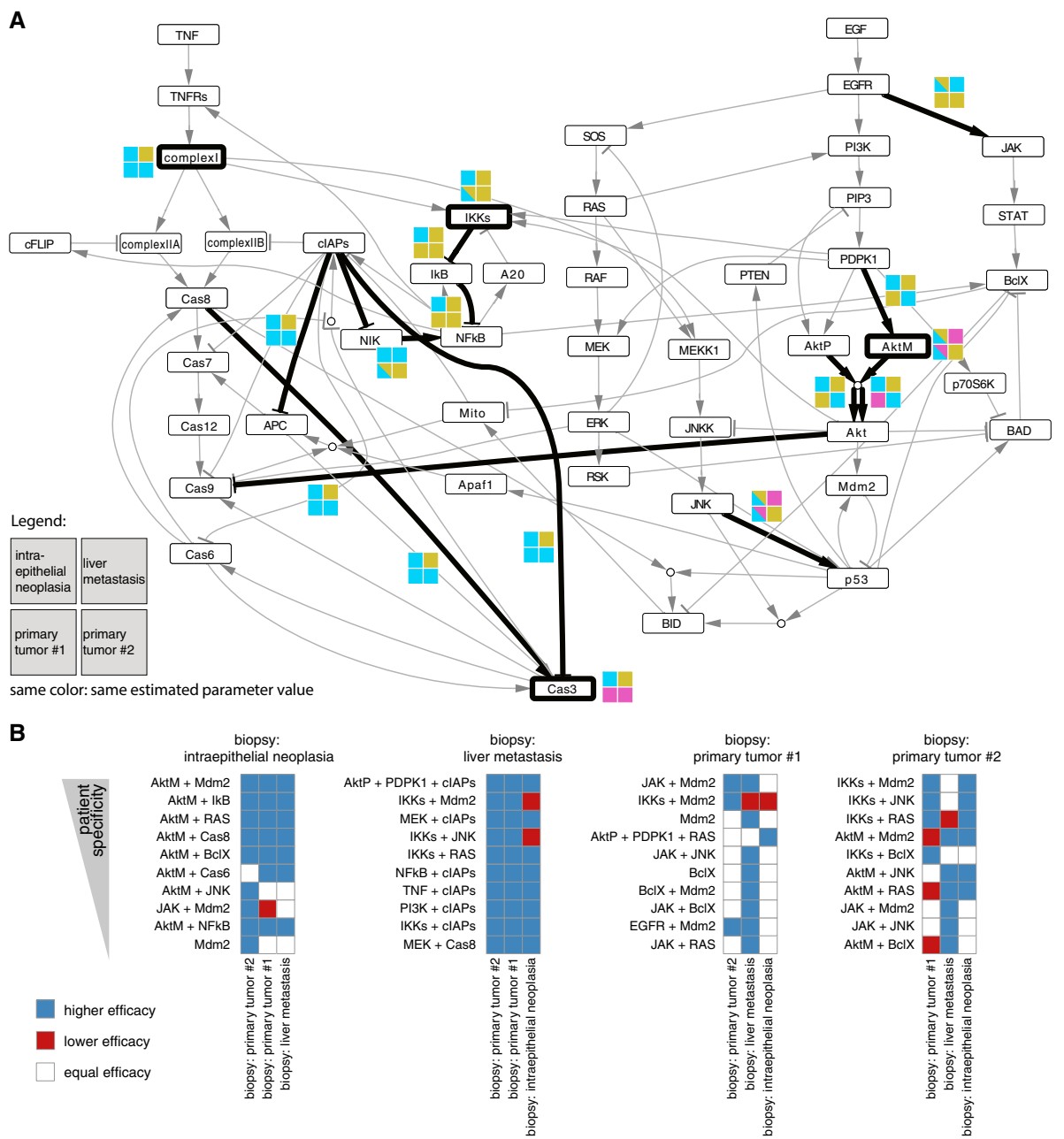

**Figure 4. Patient-specific models of signaling pathway.**

A   Mechanisms which are differentially regulated among the patients are highlighted with thick black lines. Colored squares represent the same distribution (same color) or differential distribution (different colors) across the four patient samples.

B   Patient-specific predictions of new drug combinations. Predictions are ranked for each patient and color coded to compare the efficacy with the other patients.

## Model-based prioritization of new personalized treatments

The personalized models can be used not only to investigate the patient-specific deregulated mechanisms, but also to predict novel experimental conditions as previously shown for the cell lines. For example, we can simulate the effect of a new kinase inhibitor by inhibiting the corresponding node in the model and predict its effect on Cas3 and thus on apoptosis. By implementing *in silico* testing,

we can increase the throughput of our screening method for each patient, allowing to predict the effect of new potential therapies which cannot be experimentally tested due to limited biopsy material.

For each patient, we simulated the effect of 12 new single and 174 combinatorial therapies (186 *in silico* perturbations in total) targeting nodes in our model. Having applied bootstrap when deriving our personalized optimized models, as previously described, we

obtained a distribution of the predictions for each simulated perturbation. This allowed us to perform statistical tests to compare the effect of the same treatment across patients and focus on patient-specific effects, removing 129 out of the 186 treatments that were not statistically different between patients (Kruskal–Wallis rank sum test, adjusted *P*-value < 0.01, effect size > 0.3, see Materials and Methods). With this step, we removed treatments that could not be predicted with sufficient confidence for any patient (broad distributions), and the treatments showing high predicted efficacy for all patients, that could be likely due to general toxicity. For the remaining 57 conditions, we performed also post hoc comparison between all patient pairs (Wilcoxon sum rank test, adjusted *P*-value < 0.01, effect size > 0.3, see Materials and Methods). Similar to what we noticed comparing model parameters, using these predictions we observed that primary tumors behave most similar as they show no statistical difference in 79% of the cases. This is much higher than the similarity observed in comparison with intraepithelial neoplasia (60% and 43% for primary tumors #1 and #2, respectively), or the liver metastasis (12% and 11%, respectively, for primary tumors #1 and #2).

Finally, in order to prioritize new patient-specific promising treatments, we ranked the simulated perturbations for each patient by patient specificity (i.e., higher effect size in the pairwise comparison); Fig 4B shows the top ten for each patient. Interestingly, there are three treatments showing strong potential (among the top ten) for both primary tumors. Two of these consist of targeting Mdm2 in combination with JAK and IKKs, respectively. Mdm2-p53 binding is known to be an important target in pancreatic cancer, where TP53 mutations occur in 50–70% of the patients (Morton *et al*, 2010). Finding treatments to combine with drugs disrupting this binding, like Nutlin-3 (Khoo *et al*, 2014), is currently of great interest (Bykov *et al*, 2018) especially in pancreatic cancer (Izetti *et al*, 2014). In particular, the activation of the JAK-STAT pathway has been shown to be common in pancreatic cancer (Matsuoka & Yashiro, 2016) and is often associated with TP53 mutation (Wörmann *et al*, 2016), suggesting that targeting Mdm2-p53 and JAK could be indeed a promising combination therapy for some patients. Based on our predictions, combinatorial targeting of Mdm2 and JAK is also efficacious in the intraepithelial neoplasia, while targeting Mdm2 in combination with IKKs is more efficacious in the liver metastasis.

## Discussion

Given the intrinsic complexity of cancer, experimental data obtained recording the cellular response to perturbations are essential to study cancer cells as a dynamic system. These functional data provide complementary information to that obtained by genomic profiling in steady state and are particularly relevant for investigating therapeutic efficacy of anticancer drugs (Yaffe, 2013; Letai, 2017). Further knowledge can be extracted by analysis of these experimental perturbation data via mathematical modeling, providing a rationale for mechanism-based interpretation of drug response.

We here present an approach to effectively build mechanistic models from the integration of prior knowledge of the underlying pathways and large-scale perturbation datasets that can be measured on patient samples. This integration with cell type- or patient-specific data enables a contextualization of otherwise non-specific prior knowledge. Because of the low material needed by our recently developed MPS platform (Eduati *et al*, 2018), our approach can be applied not only to *in vitro* but also to *ex vivo* settings, as demonstrated in this study. Our tool allows us to dissect functional differences in the signaling pathways by comparing the model parameters. These parameters recapitulate similarity between different tumor stages better than the drug screening data, suggesting that they can shed light on the molecular basis of tumors at the individual patient level. In addition, the tool can be used to rationally select efficacious combination therapies, as illustrated in the study in cell lines. We chose a simple logic formalism so that we could efficiently model large networks despite measuring a single readout. Importantly, the models were continuous, based on ODEs, and thus able to capture quantitative differences. *In silico* and *in vitro* analyses demonstrate that our approach is robust. In summary, our combination of mathematical modeling and *ex vivo* perturbation data helps to investigate possibly deregulated mechanisms (or pathways) and to explain specific responses to drugs directly on patients' biopsies.

The MPS platform can be extended in the future to provide richer data, and thereby improve the mathematical models. We have so far generated MPS data with a large number of perturbations but a single readout (apoptotic marker). While informative, in particular to study the effect of anticancer drugs, it is limited to capturing a subset of signaling networks. MPS can in principle be applied to other markers, as well as connected to richer output technologies, such as high-content imaging, or single-cell RNA sequencing. We expect that the breadth and depth of our models will increase when expanding readouts. In addition, our modeling approach is not limited to MPS, but can be used with different types of data describing molecular or phenotypic changes upon perturbation. Besides the readouts, extension to multi-time-point measurements will provide additional insight into the dynamics and feedback regulation of the system. Finally, a higher granularity in drug concentrations tested will provide information on intermediate effects. We expect that further developments of technologies for functional screening of cancer patient biopsies will follow in the near feature (Letai, 2017), and this will reflect in further improvement of patient-specific mathematical models that can be obtained using our pipeline.

Considering a family of models allows us to account for cell signaling uncertainty for both estimated parameter values and model predictions, which could be due to cellular heterogeneity (Kim *et al*, 2018). This is currently taken into account when comparing models and when making predictions of efficacious therapy. By using statistical tests, we consider as promising only combinations that are robustly more efficacious for each individual patient/cell line. Having few cells per plug (~100) and many replicates (at least 20 per condition), we have collected information on the heterogeneity of cellular response to drugs within a patient sample, which could be taken into account when building the model. Statistical models could be used in the future also to distinguish between the variability due to technical noise (same for all plugs) and the variability due to heterogeneity of cellular response to drugs (specific for each condition). In addition, our current approach integrates data from different cell types, providing an "average" model. Alternatively, different cell types can be sorted out prior to the MPS experiments, to obtain cell type-specific information.

Generation of perturbation data followed by mathematical modeling has proven to be a powerful tool to study cancer biology and therapies *in vitro* (Klinger *et al*, 2013; Molinelli *et al*, 2013; Eduati *et al*, 2017; Hill *et al*, 2017). The insights from *in vitro* models can be extrapolated to patient data using a patient's static profiling, such as gene expression (Fey *et al*, 2015). When no other data are available, this is certainly a very valid strategy to generate personalized models. Our work shows, however, that this basal information cannot recapitulate the insights obtained by data upon perturbation. If one can generate such data directly from patient samples, we should be able to generate more precise models that provide more accurate insights and predictions. We believe the strategy presented in this work can contribute to the development of functional precision cancer medicine.

# Materials and Methods

**Reagents and Tools table**

| Reagent/Resource | Reference or source | Identifier or catalog number |
|---|---|---|
| **Experimental Models** | | |
| *AsPC-1* | ATCC | CRL-1682 |
| BxPC-3 | ECACC | 93120816 |
| **Chemicals, enzymes and other reagents** | | |
| CellTiter-Glo 2.0 | Promega | G924C |
| Navitoclax | Selleckchem | S1001 |
| Taselisib | Selleckchem | S7103 |
| Trametinib | Selleckchem | S2673 |
| MK-2206 | Selleckchem | S1076 |
| PHT-427 | Selleckchem | S1556 |
| Gemcitabine | Selleckchem | S1149 |
| ACHP | Tocris | 4547 |
| Gefitinib | Selleckchem | S1025 |
| **Software** | | |
| *CNORode2017* | https://github.com/saezlab/CNORode2017 | |
| | Eduati *et al* (2017) | |
| **Other** | | |
| Microplate reader | Beckman-Coulter | Paradigm Detection Platform |

**Methods and Protocols**

*Microfluidics setup, screened compounds, and samples*
Data were generated using our Microfluidics Perturbation Screening (MPS) platform as presented in (Eduati *et al*, 2018). A cell suspension is generated from cell lines in culture or from patient biopsies (Fig 1A). A microfluidics chip is then used to automatically generale plugs with different chemical composition, using valves that can be opened and closed using a Braille display. In each plug, cells (about 100) are encapsulated together with one or two compounds and a rhodamine 110 (green-fluorescent dye)-based substrate of caspase-3 ((Z-DEVD)2-R110), which is a marker of apoptosis. The activation of caspase-3 causes the cleavage of the substrate and the subsequent release of the green fluorescence in the plug. Alexa Fluor 594 (orange-fluorescent dye) is added to the cell suspension to verify the proper mixing of the different components in each plug.

Samples are produced in a sequential way in multiple replicates (12 for perturbations, 20 for untreated control), and each sample is followed by a corresponding barcoding sequence produced using two different concentrations of Cascade Blue dye (blue-fluorescent dye) to encode the sample number in binary digits. The full sequence of conditions is repeated at least twice (resulting in a total of at least 24 replicates per perturbation). Aqueous plugs are separated by mineral oil plugs to avoid cross-contamination. All plugs are collected in a tube and incubated overnight for 16 h at 37°C and 5% carbon dioxide. Fluorescence in three channels (green, orange and blue) is measured for each plug by exciting it with lasers (375, 488, and 561 nm) and detecting the emissions with corresponding three photomultiplier tube (PMT) detectors (450, 521, and > 580 nm).

The 10 screened compounds (alone and in all pairwise combinations) include two cytotoxic drugs (gemcitabine and Oxaliplatin), standard of care for pancreatic cancer, 7 kinase inhibitors (ACHP: IKKi, AZD6244: MEKi, Cyt387: JAKi, GDC0941: PI3Ki, Gefitinib: EGFRi, MK-2206: AKTi, and PHT-427: AKTi & PDPK1i) and one cytokine (TNF).

In accordance with the Declaration of Helsinki of 1975, human pancreas biopsies (primary tissue samples) were obtained during routine clinical practice at University Hospital Aachen, Aachen, Germany, and were provided by the RWTH Aachen University Centralized Biomaterial Bank (cBMB) according to its regulations, following RWTH Aachen University, Medical Faculty Ethics Committee approval (decision EK 206/09). Sample processing at the EMBL in Heidelberg, Germany, was approved by the EMBL Bioethics Internal Advisory Committee. Informed consent was obtained from the patients for research use of the samples. As described in Eduati *et al* (2018), processed samples were carefully selected by the pathologist from the resected tissue to guarantee that only the specific tissue of interest was screened.

### Building the apoptosis pathway model

The logic model shown in Fig 1B was derived by manual literature curation starting from the model described by Mai and Liu (Mai & Liu, 2009) and integrating additional information in order to include all nodes perturbed in our experiments and to well describe pathway cross-talks. Logic rules were also adopted from the Boolean model of apoptosis by Schlatter *et al* (2009). We modeled both the intrinsic (mediated by the mitochondria) and the extrinsic (mediated by death receptors, TNFRs) apoptosis signals including nodes encoding both anti- and pro-apoptotic effects. Binding of TNF to TNFRs activates the extrinsic pathway mediated by caspase-8 (Cas8 in Fig 1B) activation of caspase-3 (Cas3). The two distinct caspase-8 activation pathways (Wang *et al*, 2008) are represented by the cascade involving complex I (composed of RIPK1, TRADD, TRAF2), which induces the formation of two different caspase-8 activation complexes: complex IIA (TRADD, RIPK1, FADD, Pro-caspase 8) and complex IIB (RIPK1, TRADD, FADD, Pro-caspase 8, cFLIP) that can be inhibited by cFLIP and cIAPs, respectively. For simplicity, caspase-8 is modeled as a separate node (Cas8) regulated by the two complexes. TNF can also regulate the intrinsic pathway through the activation of NFkB (anti-apoptotic node) by removal of its inhibitor IkB. The activation of the intrinsic pathway is executed by the mitochondria through the release of SMACs (second mitochondria-derived activator of caspases) and cytochrome c. The former deactivates IAPs, which are anti-apoptotic proteins, and the latter binds to Apaf1 (apoptotic protease activating factor-1) and pro-caspase 9 which is converted to its active form of caspase-9 (Cas9) and in turn activates caspase-3 (Cas3).

Both Akt and ERK have an anti-apoptotic effect by phosphorylating BAD (Balmanno & Cook, 2009) and thus unbinding it from BclX and this can be modeled as an OR gate (She *et al*, 2005). We also included the pro-apoptotic effect of ERK as regulator of p53 (Cagnol & Chambard, 2010). Additional cross-talks from RAS to MEKK1 and PI3K pathways were included as described by Grieco *et al* (2013). Additional interactions between nodes in the network were found using Omni-Path (Türei *et al*, 2016) and through manually curating the literature supporting the interactions in the databases. For example, in this way we found support for potential context-dependent cross-talks from PDPK1 to MEK (King *et al*, 2000; Borisov *et al*, 2009; Aksamitiene *et al*, 2012) and to IKK/NFkB signaling (Tanaka *et al*, 2005) which were therefore added to our prior knowledge network.

### Data normalization and formal definition of logic ODEs

Data from the MPS were preprocessed using the pipeline for data analysis and quality assessment described in Eduati *et al* (2018) and implemented in R (https://github.com/saezlab/BraDiPluS). In short:

1 We used the signal in the orange channel (see description of the Microfluidics setup) to discard corrupted data corresponding to improperly formed plugs.
2 For each screened condition in each run (i.e., full sequence of all tested conditions—see description of the microfluidics setup), we computed the median across replicates (12 plugs produced per condition, median, and standard error are reported in Appendix Fig S10) and the corresponding z-score. Additionally, we compute the FDR-corrected *P*-value with respect to the untreated control (one-sided Wilcoxon rank sum test).
3 Median *z*-score and combined *P*-values (using Fisher's method) were then computed across different runs (at least two per sample).
4 In order to be used in the logic formalisms, data were scaled between 0 (untreated control, which is also the initial state of the model) and 1 (maximum activation). Conditions which were defined as not significantly different with respect to the untreated control (combined *P*-value < 0.05) were also set to 0.

For implementing and optimizing the mathematical models, we used the CellNOptR tool (Terfve *et al*, 2012) and a modified version of the CNORode add-on to model logic-based ordinary differential equations (ODEs), as presented in Eduati *et al* (2017) and available at: https://github.com/saezlab/CNORode2017. The logic network described in the previous section "Building the apoptosis pathway model" is therefore used as a scaffold to build the logic ODE model. In the logic ODE formalism (Wittmann *et al*, 2009), each node (i.e., species $x_i$) is modeled by an ODE with a continuous update function $B_i$ representing the regulation by the $N_i$ upstream nodes.

$$\frac{\mathrm{d}x_i}{\mathrm{d}t} = \tau_i(B_i(f(x_{i1}), f(x_{i2}), \dots f(x_i N_i)) - x_i)$$

The tunable parameter $\tau_i$ represent the life-time of species *i*. We define each regulation using a sigmoidal transfer function:

$$f(x_{ij}) = 1 - \frac{\frac{(1-x_{ij})^{n_{ij}}}{(1-x_{ij})^{n_{ij}}+k_{ij}^{n_{ij}}}}{\frac{1}{1+k_{ij}^{n_{ij}}}}$$

where parameters $n_{ij}$ and $\tau_{ij}$ are fixed to 3 and 0.5, respectively, and the tunable parameter $k_{ij}$ represent the strength of the regulation of species *j* on species *i* (edge $j \rightarrow i$). When $k_{ij} = 0$, there is no regulation, which correspond to removing the interaction, while higher values of $k_{ij}$ correspond to stronger regulation. No new interactions can be created during model optimization. The effect of the compounds used to perturbed the system was simulated by forcing the activity of the specific node to 1 in case of a stimulating compound (TNF) and to 0 in case of an inhibiting compound (all kinase inhibitors).

Using the CellNoptR package, the logic model was compressed as described in Saez-Rodriguez *et al* (2009) to reduce model complexity. Parameters were estimated by fitting the model simulation to the experimental data using the optimization toolbox MEIGO (Egea *et al*,

Federica Eduati et al

Molecular Systems Biology

2014). Bootstrapped distributions for all parameters were obtained by repeating the optimization resampling data with replacement.

### Statistical tests for comparisons of parameters and predictions

Non-parametric tests were used because they are highly robust against non-normality. Pairwise comparisons (both on parameters and on predictions) for cell lines were performed using Wilcoxon rank sum test. Kruskal–Wallis rank sum test (one-way ANOVA on ranks) was used when comparing multiple groups (i.e., for patients, both on parameters and on predictions) and followed by post hoc pairwise comparison with Wilcoxon sum rank test on the parameters which are not equally distributed among all groups. Effect size w was computed for Wilcoxon rank sum test as $Z/\sqrt{N}$, where $Z$ is the statistics from the test and $N$ is the number of observations, and for the Kruskal–Wallis rank sum test, it was computed as $\sqrt{X^2/N}$ where $\chi^2$ is the statistics from the test and $N$ is the number of observations. $P$-values were always Bonferroni adjusted to correct for multiple hypothesis testing. Significance thresholds (reported also in the main text) were set to 0.01 for all adjusted $P$-values. For the effect size, the threshold was set to 0.2 when comparing model parameters and 0.3 when comparing predictions (to further limit the number of significant testable predictions). Perturbations are considered to have no effect if their median predicted value is < 0.07.

### In vitro validation experiments

AsPC1 and BxPC3 cells were used at passage 2–4 after thawing and seeded at 12,800 or 8,000 cells per well, respectively, in a 96-well plate with RPMI media (supplemented with 10% FBS, 1% penicillin/ streptomycin, 4.5 mg/ml glucose, and 1 mM sodium pyruvate) on day 0. Drugs were added on day 1, and viability was measured after 72 h using CellTiter-Glo® (Promega). Navitoclax has been used at fixed concentrations of 2.5 μM and 10 μM. Trametinib has been used at fixed concentrations of 0.25 μM and 1 μM. Taselisib and MK-2206 were used up to a concentration of 10 μM in an 8-points 1:3 dilution series, spanning a 2,000-fold range. PHT-427 was used up to a concentration of 75 μM in an 8-points 1:2 dilution series. We performed three biological replicates. For each biological replicate, each data point was measured in at least three wells per plate (i.e., technical replicates). Raw data were preprocessed by subtracting the average background (blank) and removing outliers only if one out of three technical replicates was off by > 30% compared to the other two. This resulted in a maximum removal of two data points per plate. Viability data were normalized to the negative control condition (i.e., DMSO treated cells). We fitted a four-parameter log-logistic model using the "drm" R package (Ritz et al, 2015) and computed the synergy score with Bliss independence model as implemented in the "synergyfinder" R package (He et al, 2018).

### Xenograft mouse models

Xenograft mouse experiments were performed by an external company (EPO Berlin). All animal experiments were carried out in accordance with the German Animal Welfare Act as well as the UKCCCR (United Kingdom Coordinating Committee on Cancer Research). The respective pancreas carcinoma cell suspensions of the human AsPC1 or BxPC3 cells were injected subcutaneously (s.c.) into the left flank of anaesthetized female NMRI nu/nu mice. Tumors were allowed to establish a palpable size (about 0.1 cm$^2$), before the

treatment was started. A total of 40 mice were used, grouping the animals in five groups for each cell line (four animals per group). Mice in the five groups were treated, respectively, with (i) gemcitabine (100 mg/kg i.p. once a week in PBS); (ii) combination of navitoclax (100 mg/kg p.o. Q1D) and taselisib (2.5 mg/kg p.o. Q1D); (iii) combination of trametinib (1 mg/kg p.o. Q1D) and MK-2206 (120 mg/kg p.o. sequence days 0,2,4,7,9,11); (iv) combination of navitoclax (100 mg/kg p.o. Q1D) and PHT-427 (200 mg/kg p.o. BID per 10 days); and (v) vehicle p.o. alone. During the study, tumor volumes were measured in two dimensions with a caliper. Measurements were performed twice a week to capture the growth characteristics of the different tumor models. Tumor volumes (TV) were calculated by the formula: TV = (width² × length) × 0.5. During the study, mice were maintained under sterile and controlled conditions (22°C, 50% relative humidity, 12-h light–dark cycle, autoclaved food and bedding, acidified drinking water) and monitored for body weight and health condition. Relative tumor volume (RTV) was computed with respect to day 0 (i.e., first day of treatment). Statistical comparison was performed using the R package *TumGrowth* (Enot et al, 2018) making use of linear mixed-effect models to compare longitudinal data. Holm method was used to correct for multiple comparisons.

## Data availability

Microfluidics Perturbation Screening (MPS) data are available at: https://github.com/saezlab/BraDiPluS. New validation data and the code to guarantee reproducibility of the results are available at: https://github.com/saezlab/ModelingMPS.

**Expanded View** for this article is available online.

## Acknowledgements

We thank Hayley Donnella and Attila Gabor for useful discussion and feedback on the manuscript. We thank Diana Panayotova Dimitrova and Maria Feoktistova for constructive discussion about apoptosis pathway. FE thanks European Molecular Biology Laboratory Interdisciplinary Post-Docs (EMBL EIPOD) and Marie Curie Actions (COFUND) for funding, and JRC for Computational Biomedicine which was partially funded by Bayer AG.

## Authors contributions

JS-R and CAM conceived the project. JS-R and FE designed the project. FE performed the analysis under the supervision of JS-R. PJ, MJG, JW, and TC performed experimental validation and helped with results interpretation. FE and JS-R wrote the manuscript. PJ contributed to manuscript finalization. All authors approved the final manuscript.

## Conflict of Interest

The authors declare that they have no conflict of interest.

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
