## [Review Process File · Molecular Systems Biology]

Patient-specific logic models from cancer biopsies' screening to personalize combination therapies

Federica Eduati, Patricia Jaaks, Jessica Wappler, Thorsten Cramer, Christoph Merten, Mathew Garnett and Julio Saez-Rodriguez

Review timeline:	Submission date:	20 th Sep 18
	Editorial Decision:	30 th Oct 18
	Revision received:	9 th Dec 19
	Editorial Decision:	15 th Jan 20
	Revision received:	27 th Jan 20
	Accepted:	28 th Jan 20

Editor: Maria Polychronidou

Transaction Report:

1st Editorial Decision

30th Oct 18

Thank you again for submitting your work to Molecular Systems Biology. We have now heard back from the three referees who agreed to evaluate your study. As you will see below, the reviewers acknowledge that you address a timely topic and think that the presented approach seems relevant. They raise however a series of concerns, which we would ask you to address in a major revision.

Without repeating all the points listed below, one of the more fundamental issues refers to the need to perform further experimental analyses to better support the predictive power of the proposed approach. These include experimental validations in additional cell lines as well as with further drug combinations. Moreover, it is particularly important to include further validations to strengthen the part of the study referring to the use of patient-specific models.

All other issues raised by the reviewers need to be satisfactorily addressed. As you may already know, our editorial policy allows in principle a single round of major revision. It is therefore essential to provide responses to the reviewers' comments that are as complete as possible. Please feel free to contact me in case you would like to discuss in further detail any of the issues raised by the reviewers.

REFeree REPORTS

Reviewer #1:

The manuscript by Eduati and others describes the application of what is claimed to be a novel *ex vivo* drug sensitivity screening assay to model based prediction of efficacious drug combinations. The topic is significant and almost certainly model-based approaches are required to start exploring combination space of anti-cancer drugs. The assay measures cell death response to ~dozens of anti-cancer drugs-i.e. perturbation response data. Based on such data for a particular cell line or patient

sample, they fit a logic-based ODE model that can describe the perturbation response data. They do provide some statistics regarding the fidelity of the derived models. They demonstrate that their approach is able to identify a drug combination which has more activity in one cell line vs. another. They also apply it in a forward manner to data from four different types of patient samples from pancreatic cancer, but validation of drug combination predictions here was not reported (may not have been possible). The work is definitely pushing the boundaries of creating and then using with confidence personalized models for drug combination predictions. However, I do have some substantial concerns as detailed below.

1. The approach is shown to work only in one case for two cell lines. While promising, one would ideally like to see this work in a variety of cell lines, or for more than one drug combo in pairs of cell lines. Moreover, from what this reviewer could understand from the figures, only a single (or maybe two) doses of the 2nd inhibitor of the predicted and validated combo (taselisib and navitoclax) were used. I would expect a much more robust assessment of the combination response, particularly since only a single one in two cell lines are being investigated. In addition, more discussion of the proposed mechanism of the drug combo sensitivity would be warranted (based on the model for example).
2. Similar to above, the approach is applied to build patient specific models (not from cell lines but patient samples), but the author's never show that these patient specific models have predictive power for drug combinations. That is there is a lack of validation to support the title of the paper. In my opinion, this deficiency becomes less important if more extensive cell line validation is done as above (for example).
3. Clarity regarding experiments: please confirm clearly in main text if there is only a single endpoint measurement of Caspase 3 activity as a proxy for apoptosis, and whether this is a time course.
4. Related to 3, how do the authors reconcile the fact that from tumor biopsies, many cells will not be cancer cells? Also, a single averaged measurement is used as the readout but different cells are dying at different doses. How do the author's models reason about population vs. single cell phenotypes?
5. The author's mention not including DNA damage nodes in the model, yet speak of p53 responses; more clarity is needed on how the traditional cytotoxic drugs are modeled.
6. It would be very interesting to understand more deeply which training experiments (i.e. which drug perturbations) were most important for constraining the resultant models and for making combination predictions? Sensitivity analyses of many sorts may shed light on this. The result suggesting the PHT-427 inhibitor data was most important is interesting, but why is it so?
7. The ex vivo "PBS" assay works on cells in suspension from what I could understand. Cell attachment is a large determinant of drug response, especially for most solid tumor cells. One then wonders how robust the results of this screening assay will be.
8. The author's claim that "basal information" (i.e. mutations and expression) cannot discriminate samples...only when perturbation data are included does hierarchical clustering "work" (Fig. S5). This claim seems quite undersupported (Fig. S2 is also not clear how it supports this idea). I am not sure it is a necessary claim to make, when the author's analyses are shown to make some subsequently validated predictions about drug combos.
9. With regards to modeling, it was not clear how robust the cell-type specific connection identified are. No experimental validation of these predicted links was presented. This gives rise to some other related methodological questions: how much is the initial model structure allowed to change during fitting? Can wholly new links be created? Or can only the strengths of existing links be altered?

Minor points:

1. Page 3, 2nd paragraph "allow to generate" is awkward
2. Acronym PBS is somewhat unfortunate in that it is commonly understood as phosphate buffered saline. Suggest to revise if possible at this point.
3. Clarification: pg. 6, 2nd paragraph, what does $Cas3 > 0.45$ mean?

Reviewer #2:

In the manuscript "Patient-specific logic models from cancer biopsies' screening to personalize combination therapies", Federica Eduati and coworkers derive computational models for two cancer cell lines and four biopsies. The development is based on (1) qualitative information about the pathway topology available in the literature and (2) a perturbation screening for the individual

biopsies. The perturbation screening was performed using a plug-based microfluidics platform presented in a recent manuscript. For each biopsies a logic models is calibrated, and the differences of the estimated parameters are used to suggest potential therapies. Selected predictions could be validated experimentally.

The development of truly patient specific models is highly relevant and interesting. The authors present the first model-based analysis of data obtained using a recently established plug-based microfluidics screening platform. The analysis was built upon a computational pipeline developed (and in parts published) by the authors over the last years. Overall, the manuscript is very interesting. It provides a proof of principle for the model-based analysis of biopsy-specific drug responses and assessment of dynamic differences between biopsies. The manuscript is well written and in most parts convincing. In particular the prediction and validation of drug combinations is convincing.

Major

=====

p.3. Throughout the manuscript it is not completely clear for me which data are new and which data have previously been published. Please clarify this, e.g., in the first section of the Results part.

p.4. "Parameter fitting was repeated 10 times and performances were assessed using different metrics to compare model simulations with the experimental data." - I'm wondering whether the optimizer converged. It would be interesting to see if the objective function value is the same across multiple optimizer runs.

p.5. My key concern is related to the statistical analysis. The authors use bootstrapping for uncertainty analysis and the assessment of differences between cell lines and biopsies. As parameters between cell lines / biopsies are considered different if the distribution of the differences between bootstrap samples are significant, I'm wondering what the impact of the bootstrap size is. It appears that increasing or decreasing the number of samples could have a substantial impact on the results.

In my opinion a simultaneous fitting would be more suited to determine relevant difference, see e.g. Steiert et al, L1 regularization facilitates detection of cell type-specific parameters in dynamical systems, *Bioinformatics*, i718-i726. This would also circumvent the common problem that for non-identifiable problems the bootstrapping results depend on the selected optimizers, a key problem for uncertainty analysis.

p.6. "Even when the cross-validation was repeated removing all the experiments involving a specific drug each time (instead of random ones, Figure 2C), predictions are still very good (Pearson Corr range 0.66-0.97) for all drugs except PHT-427 (Akt and PDPK1 inhibitor, Pearson Corr = -0.29). This implies that experiments with PHT-427 are essential to define the models." - I'm wondering about the negative correlation. Wouldn't it imply that the experimental data are inconsistent?

In that respect, it would also be interesting to know whether there are parameters related to particular drugs. If there exist parameters whose influence is zero unless the drug is present, I'm wondering how the authors chose them in the cross-validation.

p.8. "We here present an approach to effectively build mechanistic models" - It is not clear for me to what the authors refer to. The model building part is not really discussed in the main manuscript. Indeed, the procedure seems to be manual and without clear SOPs. In addition, the methods used for the calibration of logic models seem to be published.

I would suggest that the authors clarify the contribution. Based on the result section, it appears that the analysis of the considered perturbation data is the main result. If a novel computational approach is employed, this should in my opinion be clarified.

To ensure reproducibility and reusability, I would like to see the code and the experimental data made available.

Minor

=====

p.3. "The logic model includes AND gates (dots in Figure 1B) when all upstream regulators are

needed to activate a node" - In the figure it seems that rather circles are used instead of dots.

p.4. The variability of the metrics would be interesting.

p.5. The authors argue that the mutation and expression patterns of the cell lines cannot explain the observed differences. Do the authors have a hypothesis what could explain them?

p.6. "We then tested experimentally one of the top combinatorial therapies predicted to be highly specific for AsPC1 ..." - It is interesting to see that AsPC1 also responds at extremely low concentrations. Did the authors test which concentrations are necessary?

p.13. "Where parameters n and k are fixed to 3 and 0.5 respectively, and the tunable parameter k_{ij} represent the strength of the regulation of species j on species i (edge $j \rightarrow i$)." - It is not clear for me if k is fixed or estimated. Furthermore, n should probably be n_{ij} .

p.9. "Considering a family of models allows us to account for cell signaling heterogeneity for both estimated parameters values and model predictions (Kim et al, 2018)." - I would suggest the use the term "uncertainty" instead of "heterogeneity", otherwise the authors would in my opinion need to confirm that the distribution indeed captures the heterogeneity of the cell system.

Fig. 2E. It would be great if the authors could also plot the model prediction.

How is the initial state of the logic model selected?

As for all modeling papers, the quality of the model depends critically on the quality of the experimental data. As the authors seem to have access to 10 replicates, it would in my opinion be good to provide information about the measurement uncertainty. Complementary, it would be interesting to know whether biopsies contain mostly tumor cells or also a substantial fraction of stoma. If so, how does this influence the analysis?

Reviewer #3:

Summary

Describe your understanding of the story: In this work, a microfluidics system (PBS) was used to expose two cell lines (AsCP1 and BxPC3) and biopsy samples from four patients to different drugs that are known to interfere with apoptosis-regulating signaling pathways. The dose-dependent capacity of the drugs to elicit apoptosis was evaluated in terms of caspase-3 activation. The dose-response data were used to parametrize a Boolean model of an apoptosis-signaling network which as constructed on the basis of literature knowledge. This resulted in individualized models which account for observed differences in the apoptosis-generating efficacy of drugs by differences in the capacity of certain signaling pathways. For the two cell lines, the predictive power of individualized models for AsCP1 and BxPC3 were validated by demonstrating that the predicted differential effect of a specific drug combination (not used during model parametrization) corresponds to the experimental outcome.

What are the key conclusions: The authors conclude that their approach may contribute to highlight differences in the capacity of pro-apoptotic pathways between cell lines and human tumors from individual subjects. This way their approach may contribute to the further optimization of drug therapy of (pancreatic) tumors.

What were the methodology and model system used in this study: The authors used two cancer cell lines and biopsy samples from four patients. Screening of drug effects was performed on a plug-based screening platform (PBS). The mathematical analysis of the data was done by means of a dynamic Boolean model

General remarks

Are you convinced of the key conclusions: No, I am not convinced. To my understanding, the

Boolean model is a too drastic simplification of the true regulatory circuits underlying apoptosis. I am afraid that the parameter sets derived on the basis of such modeling technique are at the end nothing but learning the system's response by heart. In this respect, the model validation for the two cell lines is not convincing as the effect of PI3K inhibitors is per se larger in ASPC1 cells. For the more interesting case of human tumor samples, no validation of the proposed individualized models is provided at all.

Place the work in its context: Perturbation experiments using inhibitors of key regulatory proteins in signaling networks controlling growth and proliferation of cells are meanwhile standard in research aiming at the improvement of drug-based tumor therapy. The computational analysis of such data by using multivariate statistical models or Boolean network models is quite common because these approaches just require a graphical scheme of possible molecular interactions without the need to understand the true regulation of the underlying biochemical processes (see review article below). At the end, the outcome of such methods hardly exceeds the available a priori knowledge.

What is the nature of the advance (conceptual, technical, clinical): Both the experimental and computational approach are not novel; they have been already applied in related studies of this kind.

How significant is the advance compared to previous knowledge: I don't see a significant advance.

What audience will be interested in this study: Modelers in the field of systems biology practicing similar approaches.

Major points

Specific criticisms related to key conclusions. A convincing validation of the predictive power of the proposed models is lacking.

Specify experiments or analyses required to demonstrate the conclusions: In order to demonstrate the predictive capacity of the individualized models for the two cell lines, it requires more than the application of a single drug combination (Taselisib with Navitoclax). Taking advantage of the high throughput capacity of microfluidics systems it should not be a problem to check experimentally the predicted effect of drug combinations shown in Fig.2D.

Motivate your critique with relevant citations and argumentation:

Microfluidics approaches in cancer research: Ma, Y. H. V., Middleton, K., You, L. D. & Sun, Y. (2018) A review of microfluidic approaches for investigating cancer extravasation during metastasis, *Microsyst Nanoeng.* 4

Boolean models: Fumia, H. F. & Martins, M. L. (2013) Boolean Network Model for Cancer Pathways: Predicting Carcinogenesis and Targeted Therapy Outcomes, *Plos One.* 8.

Bornholdt, S. (2008) Boolean network models of cellular regulation: prospects and limitations, *J R Soc Interface.* 5, S85-S94.

Minor points

p.9 "Generation of perturbation data followed by mathematical modeling has proven to be a powerful tool" Add references supporting this statement

p. 6 "Mathematical models were then used to simulate the effect of different drug combinations acting on the pathways that were not previously tested using PBS. For each cell line we simulated the effect of 186 new perturbations (12 single drugs and 162 drug combinations)". What are the 12 new drugs? How they differ in their binding to the target protein (expressed through the parameters of the respective Boolean function) from the 10 drugs actually tested experimentally?

Overall assessment

In its present form the work cannot be recommended for publication in MSB. If the authors are able to demonstrate that a substantial number of predictions shown in Fig. 2D are in concordance with experimental data, I will be ready to reconsider the manuscript.

Reviewer #1:

The manuscript by Eduati and others describes the application of what is claimed to be a novel *ex vivo* drug sensitivity screening assay to model based prediction of efficacious drug combinations. The topic is significant and almost certainly model-based approaches are required to start exploring combination space of anti-cancer drugs. The assay measures cell death response to ~dozens of anti-cancer drugs-i.e. perturbation response data. Based on such data for a particular cell line or patient sample, they fit a logic-based ODE model that can describe the perturbation response data. They do provide some statistics regarding the fidelity of the derived models. They demonstrate that their approach is able to identify a drug combination which has more activity in one cell line vs. another. They also apply it in a forward manner to data from four different types of patient samples from pancreatic cancer, but validation of drug combination predictions here was not reported (may not have been possible). The work is definitely pushing the boundaries of creating and then using with confidence personalized models for drug combination predictions. However, I do have some substantial concerns as detailed below.

1. The approach is shown to work only in one case for two cell lines. While promising, one would ideally like to see this work in a variety of cell lines, or for more than one drug combo in pairs of cell lines. Moreover, from what this reviewer could understand from the figures, only a single (or maybe two) doses of the 2nd inhibitor of the predicted and validated combo (taselisib and navitoclax) were used. I would expect a much more robust assessment of the combination response, particularly since only a single one in two cell lines are being investigated.

We agree that further validation was required therefore, in order to address the comment of this and other reviewers, we performed additional validation experiments both *in vitro* and *in vivo*.

Our experiments are aimed at assessing if our mathematical models are able to correctly predict whether a drug combination is specific for one or another cell line. In order to achieve this, we tested *in vitro* three of the top specific drug combinations at two concentrations of the anchor drug and 8 concentrations of the library drug. Optimal concentrations were selected based on a parallel large-scale screening and represent the range of single agent responses observed in 30 pancreatic cancer cell lines. The tested concentrations allow us to compare the dose-response curve for the two cell lines and to compute the synergy score (based on Bliss independence model). For all three tested combinations the results confirmed our model predictions.

In order to verify if our model predictions also hold in an *in vivo* setting, we tested the same combinations on xenograft mouse models derived from cell lines. One of the combinations showed strong agreement with the model predictions, significantly reducing tumor growth specifically for the BxPC3 mouse models. Additionally, when compared to the standard-of-care Gemcitabine, our drug combination showed significantly better response in the BxPC3 mouse models.

We believe that overall these results confirm that our mathematical models have a strong potential for predicting personalized treatment.

The new experimental validation have been included in the Results section and the text is also reported here:

"We then tested experimentally three of the top combinatorial therapies predicted to be highly specific for one of the cell lines based on our mathematical models (Figure 3A-C). For all three cases we had concordance between model prediction and experimental validation. Interestingly, all three combinations are targeting the PI3K/Akt pathway, which is the one where most differences were highlighted between the two cell lines. As the KRAS oncogene is mutated in >95% of pancreatic adenocarcinomas (Jones et al. 2008), inhibition of signaling pathways downstream of KRAS is considered to be an attractive therapeutic approach. Against this background, we tested the combination of the Akt inhibitor MK-2206 with a MEK inhibitor (Trametinib). As predicted by our model, the validation experiments showed that the combination of Trametinib and MK-2206 is more efficacious and synergistic (based on Bliss independence model) in BxPC3 than in AsPC1 cells

(Figure 3D, Appendix Figure S6). In line with this, a recent report on preclinical models of pancreatic cancer displayed enhanced efficacy of gemcitabine plus nabPaclitaxel when combined with MK-2206 and Trametinib (Awasthi et al. 2019). KRAS-mutant colorectal cancer was found to be extremely sensitive to combined inhibition of Bcl-2/Bcl-xL and mTORC1/2 (Faber et al. 2014). Interestingly, the combination of Navitoclax (a Bcl-2/Bcl-xL/Bcl-W antagonist) with PHT-427 (targeting PDPK1 and Akt, the bona fide downstream effector of mTORC2) was predicted by our model to be specific for BxPC3 cells. Additionally, loss of function in PTEN was shown to be important for synergistic interaction between MEK and mTOR inhibitors (Milella et al. 2017). While both AsPC1 and BxPC3 are PTEN wildtype, our model identified a weaker negative feedback loop mediated by PTEN in the BxPC3 cell line that could justify the observed synergy. This prediction was also confirmed by the experimental data showing that the combination of Navitoclax and PHT-427 is more efficacious and synergistic in BxPC3 than in AsPC1 cells (Figure 3E, Appendix Figure S7). Of note, the combination of Bcl-2/Bcl-xL/Bcl-W antagonists and drugs targeting PDPK1/Akt has thus far not been discussed as a treatment option for pancreatic cancer, further demonstrating the applicability of our model to identify innovative combinatorial approaches. Finally, we tested the combination of a PI3K inhibitor (Taselisib) with the Bcl-2/Bcl-xL/Bcl-W antagonist Navitoclax. Agents targeting the PI3K pathway in combination with a Bcl-2 family inhibitor have been previously suggested to be relevant in the context of pancreatic cancer (Tan et al, 2013). Hence, being able to predict the efficacy of this combination for specific patients (or cell lines in this case) would be highly desirable. Our validation experiments showed that the combination of Taselisib and Navitoclax is more efficacious and synergistic in AsPC1 than in BxPC3 cells (Figure 3F, Appendix Figure S8), confirming our model-based predictions.

Encouraged by this in vitro results we performed in vivo validation on xenograft mouse models. As predicted by our mathematical models, when treated with a combination of Trametinib and MK-2206 both mouse models showed a significantly different response (p -value = 0.04, see Methods for details) with BxPC3-derived mice showing a stronger response (Figure 3G). For the other treatments no significant difference was found (p -value = 0.27 for Navitoclax+Taselisib, p -value = 0.22 for Navitoclax+PHT-427). Additionally, we tested the effect of Gemcitabine, the standard-of-care in pancreatic cancer, which showed a similar effect in both mouse models (p -value = 0.75). Interestingly, the combination of Trametinib and MK-2206 provides significantly better treatment with respect to Gemcitabine in the BxPC3 mouse model (p -value = 0.04), while this is not the case for the AsPC1 mouse model (p -value = 0.77; Figure 3H). Overall these results suggest that our mathematical models have a potential for predicting personalised treatment that can specifically improve in vivo response with respect to the standard-of-care."

In addition, more discussion of the proposed mechanism of the drug combo sensitivity would be warranted (based on the model for example).

We thank the reviewer for raising this point, as indeed we had not used the model to interpret the combinations. We added the following observation in the Results section:

"Interestingly, all three combinations are targeting the PI3K/Akt pathway, which is the one where most differences were highlighted between the two cell lines."

"Additionally, loss of function in PTEN was shown to be important for synergistic interaction between MEK and mTOR inhibitors (Milella et al., 2017). While both AsPC1 and BxPC3 are PTEN wildtype, our model identified a weaker negative feedback loop mediated by PTEN in the BxPC3 cell line, that could justify the observed synergy"

In addition, literature support for the proposed mechanism of drug combinations have been included in the main text, as outlined above.

2. Similar to above, the approach is applied to build patient specific models (not from cell lines but patient samples), but the author's never show that these patient specific models have predictive power for drug combinations. That is there is a lack of validation to support the title of the paper. In my opinion, this deficiency becomes less important if more extensive cell line validation is done as above (for example).

As detailed in the response above, we have now performed more extensive validation experiments including also *in vivo* validation.

3. Clarity regarding experiments: please confirm clearly in main text if there is only a single endpoint measurement of Caspase 3 activity as a proxy for apoptosis, and whether this is a time course.

We apologize to the reviewer if this was not sufficiently clear.

The full experimental procedure was described in the Methods section "Microfluidics setup, screened compounds and samples" and this aspect was also highlighted in the Discussion:

"We have so far generated PBS data with a large number of perturbations but a single readout (apoptotic marker). Besides the readouts, extension to multi-time-point measurements will provide additional insight into the dynamics and feedback regulation of the system."

We now added this detail also in the results section "Data and modeling of apoptosis pathways":

"Measurements were performed 16 hours after perturbation, when Cas3 activation was shown to reach a plateau (Eduati et al, 2018)."

4. Related to 3, how do the authors reconcile the fact that from tumor biopsies, many cells will not be cancer cells? Also, a single averaged measurement is used as the readout but different cells are dying at different doses. How do the author's models reason about population vs. single cell phenotypes?

This is a good point raised by the reviewer. Processed samples were carefully selected by the pathologist from the resected tissue to guarantee that only the specific tissue of interest was screened. They are therefore highly enriched in tumor cells, but other cells are present. We considered whether to analyse exclusively cancer cells or rather a representative collection of the cells present in the tumour, and we decided for the latter (see also Eduati et al Nat com 2018). The main reasoning for not using e.g. FACS based purification of cancer cells was to exclude selection biases for particular clones with higher expression levels while losing the ones having downregulated the marker. Furthermore, knowing that stroma cells have a significant effect on drug response of cancer cells we tried to keep a more physiological context in the plugs. As the reviewer points out, this makes the model a 'compound' or 'average' model integrating diverse cell-types. However, by having only ~100 cells/plugs and having a high number of replicates (~20), the data can give us an idea of the heterogeneity of the response of the population to a certain perturbation.

In this paper we focused on comparing cell lines and patients. We hence use the average measure and build an overall model that does not take into account the variability of the response. Instead, by performing differential analysis between cell lines/patients we focus on the strong differences that are less affected by heterogeneity, both in terms of model parameters and predictions of drug combinations.

Future development of the approach described in the paper include the development of statistical models to take into account the heterogeneity of the cellular response to drugs, performing perturbation experiments at the single cell level and/or pre-sorting different cell types. This aspect is explained in the Discussion, where we now also explicitly mentioned the averaged nature of our models:

"Considering a family of models allows us to account for cell signaling uncertainty for both estimated parameters values and model predictions, which can be due to cellular heterogeneity (Kim et al, 2018). This is currently taken into account when comparing models and when making predictions of efficacious therapy. By using statistical tests, we consider as promising only combinations that are robustly more efficacious for each individual patient/cell line. Having few cells per plug (~100) and many replicates (at least 20 per condition), we have collected information on the heterogeneity of cellular response to drugs within a patient sample, which could be taken into

account when building the model. Statistical models could be used in the future also to distinguish between the variability due to technical noise (same for all plugs) and the variability due to heterogeneity of cellular response to drugs (specific for each condition). In addition, our current approach integrates data from different cell types, providing an 'average' model. Alternatively, different cell types can be sorted out prior to the MPS experiments, to obtain cell type-specific information."

We have now included in the Methods also details about the samples preparation.

"As described in (Eduati et al, 2018), processed samples were carefully selected by the pathologist from the resected tissue to guarantee that only the specific tissue of interest was screened."

5. The author's mention not including DNA damage nodes in the model, yet speak of p53 responses; more clarity is needed on how the traditional cytotoxic drugs are modeled.

Only drugs that specifically target nodes in the network were used for model training. Chemotherapeutic drugs induce DNA damage causing the activation of p53 but they do not specifically target directly this node. For this reason the cytotoxic drugs were excluded. We attempted to explain this in the Results section "Data and modeling of apoptosis pathways":

"We incorporated in the model all nodes perturbed by specific compounds in our screening such as targeted drugs (kinase specific inhibitors) and the cytokine TNF. The effect of chemotherapeutic DNA damaging drugs was not included in the model since they inhibit DNA replication rather than acting directly on specific signaling nodes"

To make this point clearer, we add a comment specifically:

"However, nodes such as p53, which are indirectly activated by DNA damaging drugs, are included in the model since they are key elements of different pathways."

6. It would be very interesting to understand more deeply which training experiments (i.e. which drug perturbations) were most important for constraining the resultant models and for making combination predictions? Sensitivity analyses of many sorts may shed light on this. The result suggesting the PHT-427 inhibitor data was most important is interesting, but why is it so?

As highlighted when performing cross-validation leaving out all the experiments for a specific drug, the combinations involving PHT-427 are the most difficult to predict. This might be because the drug is targeting two different nodes in the network (Akt and PDPK1) that are both in the PI3K/Akt pathway. This pathway was shown to be a critical part of the network, where most of the differences between the two cell lines are concentrated.

Following the reviewer suggestion, we further investigated if any individual experiment was particularly important A) for constraining the model and B) for making combination predictions. In order to do that, we used the bootstrap optimizations, were at each iteration some of the experiments could be considered as validation set since they were left out from the training set (because of the resampling with replacement). With 500 bootstrap iterations, each individual experiment is left out between 150 and 200 times.

A) To assess if any specific experiment was essential to constrain the model parameters, we looked at the variance of the estimated parameters across all the bootstrap iterations where a specific drug combination was in the validation set. Results are shown in the new **Appendix Figure S2**. Results show that no individual experiment is particularly important to contain the parameter estimates. This is to be expected because individual drugs are used in multiple experiments (all the combinations). Therefore, even if one specific condition is missing from the training set, the model is still constrained by all the other conditions.

B) To assess if any specific experiment was essential to make predictions of drug combinations, we looked at the bootstrap iterations when the specific drug combination was in the validation set. We performed linear regression considering as input variable a matrix with 0 or 1 (depending on

whether a condition was in the training set or not), and as output vector the corresponding squared error of the prediction of the left-out condition. This analysis determined whether there are associations between some conditions being or not in the training set and the resulting prediction error. Results are shown in the new **Appendix Figure S3**, as a heatmap of the $-\log_{10}$ of the p-values only when the p-value is < 0.1 . As expected from the analysis of the parameters, also in this case there is no specific experiment that is very important for making predictions (the resulting matrix is quite sparse).

The following text was added in the Results:

*"Results from bootstrap iterations were used also to assess whether any specific experiment was essential for constraining the model (**Appendix Figure S2**) and for making predictions (**Appendix Figure S3**), by considering the left out experiments (due to resampling with replacement) as validation set. This analysis confirmed that even if a specific condition is missing from the training set, this does not significantly affect the model and the resulting predictions. This is probably due to the fact that individual drugs are used in multiple experiments (all the combinations), therefore, even if one specific condition is missing from the training set, the model is still constrained by all the other conditions."*

7. The *ex vivo* "PBS" assay works on cells in suspension from what I could understand. Cell attachment is a large determinant of drug response, especially for most solid tumor cells. One then wonders how robust the results of this screening assay will be.

This is an interesting point of discussion that we have addressed in the paper describing the novel microfluidics platform (Eduati et al., Nature Communication, 2018). Our system exploits relatively large plugs stored statically in a piece of tubing. In this setting adherent cells can form aggregates (similar to the generation of tumour spheroids) or even attach to the plug/channel interface. In previous studies (Clauell-Tormos et al., Chem Biol. 2008) we furthermore analysed the viability and proliferation behaviour of adherent cells in plugs: Even at cell densities much higher than the ones used here, full viability was maintained over several days and significant proliferation could be measured. Additionally, in order to validate our new platform, data from the microfluidics experiments were compared with more standard screening platforms and also validated *in vivo*.

The new *in vitro* and *in vivo* validations included in this paper further confirm the robustness of the experimental data measured using the microfluidics platform.

8. The author's claim that "basal information" (i.e. mutations and expression) cannot discriminate samples...only when perturbation data are included does hierarchical clustering "work" (Fig. S5). This claim seems quite undersupported (Fig. S2 is also not clear how it supports this idea). I am not sure it is a necessary claim to make, when the author's analyses are shown to make some subsequently validated predictions about drug combos.

We apologize that this was not clear: Fig. S5 (now Fig. S9) refers to the comparison of patient's clustering based, on the one hand, on the experimental perturbation data (left panel) and, on the other hand, on the resulting model parameters (right panel). Therefore it is not related to the "basal information". This figure is commented in the Results section "Personalized apoptosis models for patients' tumors" where we highlight how model parameters can recapitulate different tumor stages better than the original perturbation data that were used to train the model. We believe that this is an interesting observation since it suggests that pathway structure can help to unveil similarities between individuals that might not be straightforward when looking only at the perturbation data.

The claim on the importance of perturbation data with respect to "basal information" is instead reported in the results section "Calibration of the apoptosis model for cell lines" where we observed that there are only two differences in functional mutations (KRAS and TP53) in the pathways of interest between the two cell lines, and they would therefore not be sufficient to predict all the differences we observe when perturbing the cell lines. A similar observation can be made for the basal transcriptomics: in Fig. S2 (now Fig. S4) we looked at the differential expression between the two cell lines. As expected, differential expression did not match with difference in the signalling pathways (compared to Fig. 2A), which are driven by the functional status of the nodes.

We added a sentence in the same section to clarify this latter point:

"Differences in signaling pathways (Figure 2A) are driven by the functional status of the nodes, that cannot be inferred solely from differences in gene expression (Appendix Figure S4)."

9. With regards to modeling, it was not clear how robust the cell-type specific connection identified are. No experimental validation of these predicted links was presented. This gives rise to some other related methodological questions: how much is the initial model structure allowed to change during fitting? Can wholly new links be created? Or can only the strengths of existing links be altered?

The structure of the network was carefully manually curated with support from experts in apoptosis, in order to include all the known interactions that are relevant in the context of apoptosis to mediate response to drugs targeting nodes of interest. Therefore, we believe that our network is quite comprehensive. This is supported by the fact that the model was able to fit well the data. In the past, we had observed cases of systematic misfit of models to certain conditions and/or readouts, and we developed methods to add missing links based on this (Eduati et al Bioinformatics 2012; Gjerga et al FOSBE 2019). In this case, however, there is no experimental support for the addition of links, hinting that the network is fairly complete.

Besides altering the strength of links, these can be also removed. Specifically, the different mechanisms of interaction were modeled using AND and OR gates as described in the Methods section "Building the apoptosis pathway model". This network was then used as general scaffold for the logic ordinary differential model as described in "Data normalisation and formal definition of logic ODEs". Model parameters represent the strength of the regulatory interactions and are tuned when models are trained on the experimental data. Setting the parameter to zero corresponds to removing the corresponding interaction.

We now better clarify these concepts adding the following sentences in the Methods section "Data normalisation and formal definition of logic ODEs":

"The logic network described in the previous section "Building the apoptosis pathway model" is therefore used as a scaffold to build the logic ODE model." ... (description of model functions and parameters) ... "When $k_{ij} = 0$ there is no regulation, which correspond to removing the interaction, while higher values of k_{ij} correspond to stronger regulation. No new interactions can be created during model optimization."

Performing a bootstrap analysis we can assess the robustness of the different interactions by looking at the confidence intervals of the corresponding estimated parameters. In the revised version of the paper we include this information in **Appendix Table S1**.

Minor points:

1. Page 3, 2nd paragraph "allow to generate" is awkward

Thanks. We rephrased it as "With recently developed organoid technologies it became possible to generate..."

2. Acronym PBS is somewhat unfortunate in that it is commonly understood as phosphate buffered saline. Suggest to revise if possible at this point.

We agree with the reviewer that this acronym can be confusing and we now use Microfluidics Perturbation Screenings (MPS) instead.

3. Clarification: pg. 6, 2nd paragraph, what does $Cas3 > 0.45$ mean?

Model predictions are between 0 and 1. We now modified the text and also the way of visualizing the model predictions in Fig. 2D and Fig S5 (we do not use a cut-off on the predicted values anymore). The new text is as follows:

"In particular, we identified the predictions which were significantly different between the two cell lines (Wilcoxon sum rank test, adjusted p-value <0.01, effect size > 0.3, see Methods; Appendix Figure S5), focusing in on those that involve the inhibition of pairwise combinations of the 12 targetable nodes (JAK, MEK, RAS, NFkB, JNK, TNF, AktM, Mdm2, EGFR, PI3K, BclX, IKKs; Figure 2D). This results in 66 drug combinations, 34 of which are specific for AsPC1 and 13 for BxPC3 and 10 have no effect on both cell lines."

Reviewer #2:

In the manuscript "Patient-specific logic models from cancer biopsies' screening to personalize combination therapies", Federica Eduati and coworkers derive computational models for two cancer cell lines and four biopsies. The development is based on (1) qualitative information about the pathway topology available in the literature and (2) a perturbation screening for the individual biopsies. The perturbation screening was performed using a plug-based microfluidics platform presented in a recent manuscript. For each biopsies a logic models is calibrated, and the differences of the estimated parameters are used to suggest potential therapies. Selected predictions could be validated experimentally.

The development of truly patient specific models is highly relevant and interesting. The authors present the first model-based analysis of data obtained using a recently established plug-based microfluidics screening platform. The analysis was built upon a computational pipeline developed (and in parts published) by the authors over the last years. Overall, the manuscript is very interesting. It provides a proof of principle for the model-based analysis of biopsy-specific drug responses and assessment of dynamic differences between biopsies. The manuscript is well written and in most parts convincing. In particular the prediction and validation of drug combinations is convincing.

We thank the reviewer for the positive comment.

Major

=====

p.3. Throughout the manuscript it is not completely clear for me which data are new and which data have previously been published. Please clarify this, e.g., in the first section of the Results part.

We apologize that this was not clear. All the microfluidics data are from Eduati et al, Nature Communication, 2018 while all the validation data are new. We now clarify it in the second paragraph of the Results section "Data and modeling of apoptosis pathways" as suggested by the reviewer.

"Models were then trained using the patient-specific experimental data from (Eduati et al, 2018) to obtain personalized models."

p.4. "Parameter fitting was repeated 10 times and performances were assessed using different metrics to compare model simulations with the experimental data." - I'm wondering whether the optimizer converged. It would be interesting to see if the objective function value is the same across multiple optimizer runs.

The barplot of the mean squared error (MSE; which is the objective function) with mean and standard deviation across the 10 runs is shown in Appendix Figure S1A.

p.5. My key concern is related to the statistical analysis. The authors use bootstrapping for uncertainty analysis and the assessment of differences between cell lines and biopsies. As parameters between cell lines / biopsies are considered different if the distribution of the differences between bootstrap samples are significant, I'm wondering what the impact of the bootstrap size is. It

appears that increasing or decreasing the number of samples could have a substantial impact on the results.

This is a valid concern by the reviewer. To address it, we tested the effect of changing the size of the bootstrap on the results of the statistical analysis for comparison of the resulting models for AsPC1 and BxPC3. As shown in the figure below (Figure R1), the results tends to stabilize around 200 bootstrap repetitions for the effect size, which is the limiting factor when selecting the significantly different parameters. Therefore having, e.g. 300 or 500 bootstrap repetitions does not affect the results and the 500 bootstrap repetitions we used in the paper are more than sufficient to have stable results.

Figure R1. A) Effect size and B) p-value of the statistical comparisons of AsPC1 and BxPC3 models for increasing bootstrap sizes. Parameters that were selected as significantly different between the two cell lines based on the 500 bootstrap repetitions are marked in orange.

In my opinion a simultaneous fitting would be more suited to determine relevant difference, see e.g. Steiert et al, L1 regularization facilitates detection of cell type-specific parameters in dynamical systems, *Bioinformatics*, i718-i726. This would also circumvent the common problem that for non-identifiable problems the bootstrapping results depend on the selected optimizers, a key problem for uncertainty analysis.

The reviewer poses a valid question. The work of Steiert et al and others is very valuable and indeed related to ours. We think that for our case, however, our approach is adequate: the approach presented in Steiert et al. allows the comparison of only two cell types since the regularization term is based on the log fold changes between the two, while we aim at comparing across multiple cell

lines or patients. Moreover, simultaneous fitting as proposed in Steiert et al. was applied to a relatively small problem (each model has ~ 10 parameters) and scalability to much bigger networks would be very challenging if feasible at all (we have 93 parameters, the simultaneous fitting of 2 cell lines and 4 patients would require fitting 558 parameters).

Additionally, our aim is to define personalized models with the end goal of having a tool that can be used in the clinics to prioritize personalized treatment. While simultaneous fitting might improve the parameter estimates from a computational perspective, it would strongly limit the applicability in a clinical setting because models for all patients would need to be optimized simultaneously and the addition of a new patient would affect all the estimates.

Our tool CNORode2017, which we used in this paper, also allows to include L1 regularization to improve parameter estimates by inducing sparsity in the network, as we have previously shown in Eduati et al., Cancer Research, 2017. However, in this paper we decided to set the regularization parameter to zero because it did not have an effect neither on predictions nor on parameter values. We believe that this is likely because the starting network was manually curated specifically for this problem.

In our case we use the same optimizer for all models and we focus the analysis on the parameters that have significantly different distributions based on bootstrap, therefore we circumvent the problem mentioned by the reviewer.

p.6. "Even when the cross-validation was repeated removing all the experiments involving a specific drug each time (instead of random ones, Figure 2C), predictions are still very good (Pearson Corr range 0.66-0.97) for all drugs except PHT-427 (Akt and PDPK1 inhibitor, Pearson Corr = -0.29). This implies that experiments with PHT-427 are essential to define the models." - I'm wondering about the negative correlation. Wouldn't it imply that the experimental data are inconsistent?

This is a good point raised by the reviewer. Indeed, it could be the case. However, the negative correlation corresponding to the PHT-427 is not statistically significant (p -value = 0.488) suggesting that this is just noise. We have now added all the corresponding p -values in Fig. 2C.

In that respect, it would also be interesting to know whether there are parameters related to particular drugs. If there exist parameters whose influence is zero unless the drug is present, I'm wondering how the authors chose them in the cross-validation.

First of all, we would like to specify that the parameters for the main analysis in the paper were chosen using bootstrap on the whole dataset rather than from the cross-validation. This is described in the section "Calibration of the apoptosis model for cell lines". Cross-validation was then used in the following section ("Model predictions and validation") as a first way of assessing the predictive power of our modeling approach: in this case models were re-optimized but using only part of the data for the training and part for the testing.

Additionally, following the suggestion of the reviewer (similar comment was raised also by Reviewer #1) we now further investigate if any individual experiment was particularly important A) for constraining the model and B) for making combination predictions. The performed analysis and the corresponding addition to the manuscript are detailed in the response to comment 6 of Reviewer #1.

p.8. "We here present an approach to effectively build mechanistic models" - It is not clear for me to what the authors refer to. The model building part is not really discussed in the main manuscript. Indeed, the procedure seems to be manual and without clear SOPs. In addition, the methods used for the calibration of logic models seem to be published. I would suggest that the authors clarify the contribution. Based on the result section, it appears that the analysis of the considered perturbation data is the main result. If a novel computational approach is employed, this should in my opinion be clarified.

We apologize for the lack of clarity. The main novelty is indeed the adaptation and application of our existing modeling framework to perturbation data that can be measured on patient samples. This

allowed us to generate patient-specific dynamic models (to our knowledge, for the first time from solid tissues) that could be used to predict new therapeutic strategies that were validated *in vitro* and *in vivo*. We now rephrased the sentence as follows:

"We here present an approach to effectively build mechanistic models from the integration of prior knowledge of the underlying pathways and large scale perturbation datasets that can be measured on patient samples. This integration with cell-type or patient-specific data enables a contextualization of otherwise non-specific prior knowledge. Because of the low material needed by our recently developed MPS platform (Eduati et al, 2018), our approach can be applied not only to in vitro but also to ex vivo settings, as demonstrated in this study."

The reviewer is right that in this study the model building was largely manual and we did not provide an SOP. We now provide all the code of our analyses in a github repository (<https://github.com/eduati/ModelingMPS>), for further use by the community. The code can be easily adapted to other contexts and datasets. The set up of the network can be done automatically by retrieving the information from knowledge databases. For this, we offer Omnipath, that provides one-point access to multiple highly-curated pathway resources. Omnipath has a functionality that, given a set of drugs, find their protein targets (through ChEMBL), and then curated causal interactions among them. We include in our repository a workflow using OmniPath for model building.

To ensure reproducibility and reusability, I would like to see the code and the experimental data made available.

We agree with the reviewer that experiments and code should be available to ensure reproducibility and reusability. We make the code and the experimental data available at <https://github.com/eduati/ModelingMPS>

Minor

p.3. "The logic model includes AND gates (dots in Figure 1B) when all upstream regulators are needed to activate a node" - In the figure it seems that rather circles are used instead of dots.

Thank, we changed it.

p.4. The variability of the metrics would be interesting.

The variability of the evaluation metrics across 10 runs is shown in Fig. S1.

p.5. The authors argue that the mutation and expression patterns of the cell lines cannot explain the observed differences. Do the authors have a hypothesis what could explain them?

We believe that this is due to the fact that pathway dynamics are driven by the functional status of the nodes. Static information such as mutation and gene expression are not sufficient to infer the difference in the dynamic of the system. We (and others, e.g. Letai, Nature Medicine, 2017) argue that cells are complex systems and, as such, their behaviour is very difficult to predict only from the initial conditions. We (e.g. Eduati et al Cancer Res 2017) and others have observed this and relatedly how static information have poor predictability of the effect of drugs.

We have now added a sentence in the Results section "Calibration of the apoptosis model for cell lines" to further clarify this.

"Differences in signaling pathways (Figure 2A) are driven by the functional status of the nodes, that cannot be inferred solely from differences in gene expression (Appendix Figure S4)."

p.6. "We then tested experimentally one of the top combinatorial therapies predicted to be highly specific for AsPC1 ..." - It is interesting to see that AsPC1 also responds at extremely low concentrations. Did the authors test which concentrations are necessary?

We are not sure if we understand the question of the reviewer correctly but indeed, a range of concentrations were tested for all the *in vitro* validation experiments, to ascertain the necessary concentration for each drug combination.

p.13. "Where parameters n and k are fixed to 3 and 0.5 respectively, and the tunable parameter k_{ij} represent the strength of the regulation of species j on species i (edge $j \rightarrow i$)." - It is not clear for me if k is fixed or estimated. Furthermore, n should probably be n_{ij} .

We apologize for the mistake, the correct notation is:

"parameters n_{ij} and τ_{ij} are fixed to 3 and 0.5 respectively, and the tunable parameter k_{ij} represent the strength of the regulation of species j on species i (edge $j \rightarrow i$)." "

p.9. "Considering a family of models allows us to account for cell signaling heterogeneity for both estimated parameters values and model predictions (Kim et al, 2018)." - I would suggest the use the term "uncertainty" instead of "heterogeneity", otherwise the authors would in my opinion need to confirm that the distribution indeed captures the heterogeneity of the cell system.

We agree with the reviewed on the suggested terminology. The assumption that the uncertainty might be due to heterogeneity was made in the cited paper. We have now rephrased as follows:

"Considering a family of models allows us to account for cell signaling uncertainty for both estimated parameters values and model predictions, which could be due to cellular heterogeneity (Kim et al, 2018)." "

Fig. 2E. It would be great if the authors could also plot the model prediction.

The model predictions for the three validated combinations are now shown in Fig. 3A-C.

How is the initial state of the logic model selected?

As described in the Methods section "Data normalisation and formal definition of logic ODEs" the data are scaled so that the untreated control is 0. Since the initial state of the model corresponds to the untreated condition, this is always set to 0. We now rephrased it to make it more clear:

"In order to be used in the logic formalisms, data were scaled between 0 (untreated control, which is also the initial state of the model) and 1 (maximum activation)." "

As for all modeling papers, the quality of the model depends critically on the quality of the experimental data. As the authors seem to have access to 10 replicates, it would in my opinion be good to provide information about the measurement uncertainty.

We agree with the reviewer that the information on the measurement uncertainty is relevant for this manuscript therefore we added as **Appendix Figure S10**.

Complementary, it would be interesting to know whether biopsies contain mostly tumor cells or also a substantial fraction of stoma. If so, how does this influence the analysis?

This is an important point, also raised by Reviewer #1. As we discussed in the response to her/his comment (point 4), the samples are carefully selected by the pathologist from the resected tissue so as to be enriched in cancer cells, but there are other cells (tissue sections and clinical data of the patient biopsies are provided as supplementary information in Eduati et al., Nat Commun, 2018).

Hence, our models represent a 'bulk' or compound model of the present cells, and this has to be kept in mind when interpreting them. For the specific aim of finding combinations to kill cancer, this is less of a concern, as most cells are cancerous. This is an area we plan to further study in the future, including the generation of data and subsequent modeling for sorted cell types.

Reviewer #3:

Summary

Describe your understanding of the story: In this work, a microfluidics system (PBS) was used to expose two cell lines (AsCP1 and BxPC3) and biopsy samples from four patients to different drugs that are known to interfere with apoptosis-regulating signaling pathways. The dose-dependent capacity of the drugs to elicit apoptosis was evaluated in terms of caspase-3 activation. The dose-response data were used to parametrize a Boolean model of an apoptosis-signaling network which was constructed on the basis of literature knowledge. This resulted in individualized models which account for observed differences in the apoptosis-generating efficacy of drugs by differences in the capacity of certain signaling pathways. For the two cell lines, the predictive power of individualized models for AsCP1 and BxPC3 were validated by demonstrating that the predicted differential effect of a specific drug combination (not used during model parametrization) corresponds to the experimental outcome.

What are the key conclusions: The authors conclude that their approach may contribute to highlight differences in the capacity of pro-apoptotic pathways between cell lines and human tumors from individual subjects. This way their approach may contribute to the further optimization of drug therapy of (pancreatic) tumors.

What were the methodology and model system used in this study: The authors used two cancer cell lines and biopsy samples from four patients. Screening of drug effects was performed on a plug-based screening platform (PBS). The mathematical analysis of the data was done by means of a dynamic Boolean model

General remarks

Are you convinced of the key conclusions: No, I am not convinced. To my understanding, the Boolean model is a too drastic simplification of the true regulatory circuits underlying apoptosis.

We agree a Boolean model would be a drastic simplification, and in fact our models are not Boolean but continuous. Our models are based on a logic formalism, from which one would indeed typically derive a Boolean model (ie. with binary ‘ON/OFF’ variables). But in our case, we derive ordinary differential equations (ODEs) from the logic network. That is, our models are actually not Boolean, but have variables that are continuous (between 0 and 1), and are also continuous in time.

Accordingly, the model contains continuous parameters, that are fitted to the data. At the same time, by describing the molecular processes as causal links instead of biochemical reactions, we are able to efficiently model a large signaling network.

This was explained in the Results (section Data and modeling of apoptosis pathways, 3rd paragraph) and Methods (section Data normalisation and formal definition of logic ODEs). We have now further clarified this in the text in the introduction:

“We obtained cell-line and patient specific continuous logic models based on Ordinary Differential Equations (ODEs).”

Results (Data and modeling of apoptosis pathways):

“Thus, they are not limited to capture only binary events as is the case for Boolean models.”

And Discussion:

“Importantly, the models were continuous, based on ODEs, and thus able to capture quantitative differences.”

I am afraid that the parameter sets derived on the basis of such modeling technique are at the end nothing but learning the system's response by heart. In this respect, the model validation for the two cell lines is not convincing as the effect of PI3K inhibitors is per se larger in ASPC1 cells.

We provide now further validations of predictions of our models, that support that our models are not just learning the system's response by heart. The new validation is presented in the Results section "Model predictions and validation" and is described also in response to Reviewer #1 comment 1.

For the more interesting case of human tumor samples, no validation of the proposed individualized models is provided at all.

We agree that the most interesting results are those obtained directly with patients. However, it is neither legally possible nor ethically acceptable to perform validation on patients. We instead performed validations with xenograft mouse models (see Results section "Model predictions and validation" and response to Reviewer #1 comment 1).

That said, we believe that our approach can eventually be of help directly for patients, by providing information that oncologist can take into account when making decisions about the treatment.

Place the work in its context: Perturbation experiments using inhibitors of key regulatory proteins in signaling networks controlling growth and proliferation of cells are meanwhile standard in research aiming at the improvement of drug-based tumor therapy. The computational analysis of such data by using multivariate statistical models or Boolean network models is quite common because these approaches just require a graphical scheme of possible molecular interactions without the need to understand the true regulation of the underlying biochemical processes (see review article below).

We agree that perturbation experiments are a common approach to study signaling networks, and also to model them using diverse modeling approaches including multivariate statistics and Boolean networks. We and others have indeed applied this, in particular *in vitro* with cell lines. What is novel in our case is the use of data derived directly from tissues from patients, which allow us to generate patient-specific dynamic models (to our knowledge, for the first time from solid tissues), and to use these models to predict combination therapies that are tested *in vitro* and, in this revised version, also *in vivo* in mouse models.

At the end, the outcome of such methods hardly exceeds the available a priori knowledge.

While the prior knowledge is used as a 'scaffold' of potential interactions among molecules, we are able to identify patient specific insights in terms of pathway wiring by fitting to patient-specific data. This is shown by the differences in the models, and by the fact that we find predictions that are patient-specific. This specific knowledge is not provided by the prior knowledge, but is essential to understand why certain patients respond to treatments while others do not.

We have added a sentence in the discussion to emphasize this point:

"This integration with cell-type or patient-specific data enables a contextualization of otherwise non-specific prior knowledge."

What is the nature of the advance (conceptual, technical, clinical): Both the experimental and computational approach are not novel; they have been already applied in related studies of this kind. How significant is the advance compared to previous knowledge: I don't see a significant advance.

As stated above, the novelty lies in the use of dynamic modeling (in our case logic-ODEs) to patient-derived perturbation data. At the onset of this project, it was not even clear if we could fit effectively such models given the nature of the data: large number of perturbations with a single readout, while our previous work (and that of others) had used more readouts (but typically fewer perturbations). It was hence already an important advance for us to be able to generate robust models from this data - we would argue this is (at least technically) an important result.

Further, by demonstrating that these models can predict drug combinations, and given that we can generate this data directly from clinical samples, we believe that our approach has a potential application in the mid-term for personalized medicine directly on patients.

What audience will be interested in this study: Modelers in the field of systems biology practicing similar approaches.

We agree that our studies will be of interest to modellers. We believe that it will also be relevant for experimental researchers, as it shows how screening data can be used to build network models that uncover mechanistic insights. We think that it will also be attractive to those more interested in medical and clinical applications, as we show how these models can be used to propose therapeutic interventions.

Major points

Specific criticisms related to key conclusions. A convincing validation of the predictive power of the proposed models is lacking.

Specify experiments or analyses required to demonstrate the conclusions: In order to demonstrate the predictive capacity of the individualized models for the two cell lines, it requires more than the application of a single drug combination (Taselisib with Navitoclax). Taking advantage of the high throughput capacity of microfluidics systems it should not be a problem to check experimentally the predicted effect of drug combinations shown in Fig.2D.

We have now extended significantly the validation, both *in vitro* and *in vivo* with mouse xenografts. We have used an independent technology for the validation because we consider that this provides a more robust support to our findings. The new validation experiments and the corresponding modified text are described in the response to Reviewer #1 comment 1.

Motivate your critique with relevant citations and argumentation:

Microfluidics approaches in cancer research: Ma, Y. H. V., Middleton, K., You, L. D. & Sun, Y. (2018) A review of microfluidic approaches for investigating cancer extravasation during metastasis, *Microsyst Nanoeng.* 4

Boolean models: Fumia, H. F. & Martins, M. L. (2013) Boolean Network Model for Cancer Pathways: Predicting Carcinogenesis and Targeted Therapy Outcomes, *Plos One.* 8.

Bornholdt, S. (2008) Boolean network models of cellular regulation: prospects and limitations, *J R Soc Interface.* 5, S85-S94.

These and other studies are related to ours in that they use microfluidics (Ma et al) and logic modeling (Fumia and Martins, Bornholdt; Boolean in their cases) to study cancer. As discussed above, we believe that the novelty of our approach relies on using microfluidic-derived perturbation data to reconstruct patient-specific logic models, and to use these to understand cancer deregulation, and explore strategies to treat it.

Minor points

p.9 "Generation of perturbation data followed by mathematical modeling has proven to be a powerful tool" Add references supporting this statement

We have added the following reference to that point:

Molinelli EJ, Korkut A, Wang W, Miller ML, Gauthier NP, Jing X, Kaushik P, He Q, Mills G, Solit DB, Pratilas CA, Weigt M, Braunstein A, Pagnani A, Zecchina R & Sander C (2013) Perturbation biology: inferring signaling networks in cellular systems. *PLoS Comput. Biol.* **9**: e1003290

We also cite here the following papers, already cited elsewhere in our manuscript:

Eduati F, Doldan-Martelli V, Klinger B & Cokelaer T (2017) Drug resistance mechanisms in colorectal cancer dissected with cell type-specific dynamic logic models. *Cancer Res.* **77**: 3364–337

Hill SM, Nesser NK, Johnson-Camacho K, Jeffress M, Johnson A, Boniface C, Spencer SEF, Lu Y, Heiser LM, Lawrence Y, Pande NT, Korkola JE, Gray JW, Mills GB, Mukherjee S & Spellman PT (2017) Context Specificity in Causal Signaling Networks Revealed by Phosphoprotein Profiling. *Cell Syst* **4**: 73–83.e10

Klinger B, Sieber A, Fritsche-Guenther R, Witzel F, Berry L, Schumacher D, Yan Y, Durek P, Merchant M, Schäfer R, Sers C & Blüthgen N (2013) Network quantification of EGFR signaling unveils potential for targeted combination therapy. *Mol. Syst. Biol.* **9**: 673

p. 6 "Mathematical models were then used to simulate the effect of different drug combinations acting on the pathways that were not previously tested using PBS. For each cell line we simulated the effect of 186 new perturbations (12 single drugs and 162 drug combinations)". What are the 12 new drugs? How they differ in their binding to the target protein (expressed through the parameters of the respective Boolean function) from the 10 drugs actually tested experimentally?

In this case we are not referring to any specific drug, but rather to hypothetical drugs that could target the nodes (proteins) in the model, in order to identify possibly relevant targets. For the interesting cases, we then followed up with validation experiments using actual drugs to target those nodes of interest as described in the Results "Model predictions and validation". The effect of inhibitors on nodes is modeled by setting the activity of the specific node to zero, therefore deactivating the specific node, as most drugs (e.g. kinase inhibitors) do.

We apologize that this was not properly explained in the previous version of the manuscript and we have now updated the text in the Results "Model prediction and validation":

"Mathematical models were then used to simulate the effect of targeting all nodes of the network in pairwise combinations, therefore simulating the effect of potential new drugs acting on the pathways that were not previously tested using MPS. For each cell line we simulated the effect of 186 new perturbations (12 on individual nodes and 162 on node pairs), by inhibiting the corresponding node in the model."

And in the Methods:

"The effect of the compounds used to perturb the system was simulated by forcing the activity of the specific node to 1 in case of a stimulating compound (TNF), and to 0 in case of an inhibiting compound (all kinase inhibitors)."

Overall assessment

In its present form the work cannot be recommended for publication in MSB. If the authors are able to demonstrate that a substantial number of predictions shown in Fig. 2D are in concordance with experimental data, I will be ready to reconsider the manuscript.

This is a valid request. As stated above, we have now added additional *in vitro* and *in vivo* validations. Through this and the further changes, we hope that the reviewer considers the work now suitable for publication.

2nd Editorial Decision

15th Jan 20

Thank you again for submitting your revised work to Molecular Systems Biology. We have now heard back reviewer #1 who agreed to evaluate your study. As you will see below, reviewer #1 is satisfied with the performed revisions and is supportive of publication.

Before we formally accept your study for publication, we would ask you to address a few remaining editorial issues listed below.

REFEREE REPORTS

Reviewer #1:

The revised manuscript does an adequate job of addressing the more substantial concerns raised in the first round of review, and the additional experiments are well received. This will be an excellent contribution to an important but very difficult area of matching drug combinations to particular cancer patients.

Corresponding Author Name: Julio Saez-Rodriguez

Manuscript Number: Molecular systems biology

Manuscript Number: MSB-18-8664R